# Reduced neural activity but improved coding in rodent higher-order visual cortex during locomotion

Amelia J. Christensen[1✉] & Jonathan W. Pillow [2]

Running profoundly alters stimulus-response properties in mouse primary visual cortex (V1), but its effect in higher-order visual cortex is under-explored. Here we systematically investigate how visual responses vary with locomotive state across six visual areas and three cortical layers using a massive dataset from the Allen Brain Institute. Although previous work has shown running speed to be positively correlated with neural activity in V1, here we show that the sign of correlations between speed and neural activity varies across extra-striate cortex, and is even negative in anterior extra-striate cortex. Nevertheless, across all visual cortices, neural responses can be decoded more accurately during running than during stationary periods. We show that this effect is not attributable to changes in population activity structure, and propose that it instead arises from an increase in reliability of single-neuron responses during locomotion.

[1] Department of Neuroscience, Washington University, St. Louis, MO, USA. [2] Princeton Neuroscience Institute & Department of Psychology, Princeton University, Princeton, NJ, USA. ✉email: pillow@princeton.edu

To understand perception, it is important to study how contextual variables (like behavioral state) affect the representation of sensory information in neural populations. Locomotion, a highly ethological behavior in rodents, is accompanied by pronounced changes in the magnitude and consistency of neural responses to visual stimuli[1–9]. For example, in mouse V1, firing rates increase[2], response variability decreases[4,9], noise correlations decrease[8], and signal-to-noise ratio (SNR) increases[8] during bouts of running. These observations have led to a prevalent view in which running acts to enhance visual representations both in activity level and coding accuracy[1–4,8,10]. Here we used the Allen Institute Brain Observatory[11] dataset to quantify how running speed affects visual responses in six visual cortical regions, evaluating hundreds to thousands of cells in each region and cortical layer. Our results highlight a dissociation between changes in activity level and coding accuracy in different locomotive states and visual regions.

## Results

**Single neuron running-speed tuning.** We evaluated locomotion tuning in primary visual cortex (V1), lateral visual cortex (VISl) ('LM', or secondary visual cortex), posterior medial visual cortex (VISpm) (a putative ventral stream region[12]), anterior lateral visual cortex (VISal), anterior medial visual cortex (VISam), and

rostral lateral visual cortex (VISrl) (putative dorsal stream regions[12]), in cortical layers 2/3, 4, and 5 (Fig. 1a). In all regions, there existed a diversity of tuning to running speed, including neurons whose activity (as measured by normalized Ca2+ fluorescence) was positively correlated with running, neurons whose activity was negatively correlated with running speed, and neurons with significant but non-monotonic correlation with running speed (Fig. 1b). A substantial fraction of neurons in all regions and layers were tuned to running speed, however more neurons were significantly modulated during running when the modulation was calculated during stimulus presentation (with all stimuli in the allen institute dataset – natural movies, natural images, drifting gratings, static gratings, and locally sparse noise – included), than when correlation was calculated in the absence of a visual stimulus (Fig. 1c, d). This was consistent regardless of whether we considered only natural stimuli (natural movies, natural images) or artificial stimuli (drifting gratings, static gratings, locally sparse noise) (Supplemental Fig. 1f, g).

However, surprisingly, we found that in some higher order visual cortices (especially VISam, VISpm, and VISrl), increased running speed was often correlated with *decreased* neural activity, as indicated by negative average neural correlation coefficients (Fig. 1e, f). The transition from net positive correlation to running to net negative correlation to running progressed along

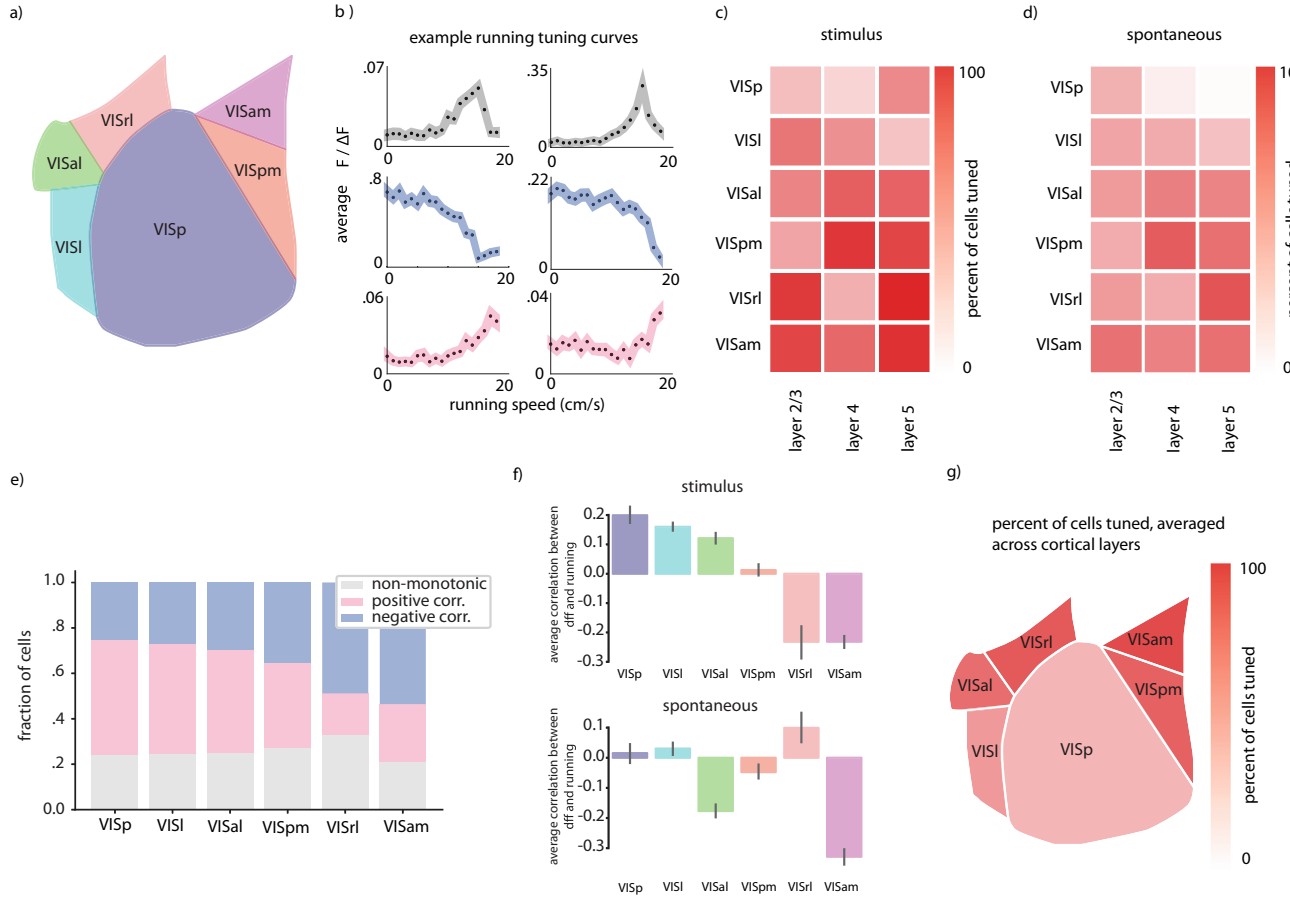

**Fig. 1 Net correlation between neural activity and locomotion varies widely across different extrastriate regions. a** Schematic of regions included in study. **b** Example non-monotonic, monotonically decreasing, and monotonically increasing tuning curves for running speed, top to bottom. Example cells are VISp layer 2/3. Error envelope represents standard error margin. **c** Overall fraction of neurons significantly tuned to running in each region and Cre-line, calculated during visual stimulus presentation. **d** Same as (**c**) except calculated in the absence of visual stimuli. **e** Fraction of cells displaying different tuning types to running speed across the visual regions we examined. **f** Average correlation coefficient between neural activity and running in each visual region, amongst neurons displaying significant ($p < 0.05$) monotonic tuning to running. Error bars are 95% confidence interval for the mean estimate calculated via 1000 bootstraps. **g** Visualization of spatial distribution of overall tuning to running in the visual regions. Cell numbers included in each analysis can be found in Supplementary Fig. 1. Source data are provided as a Source Data file.

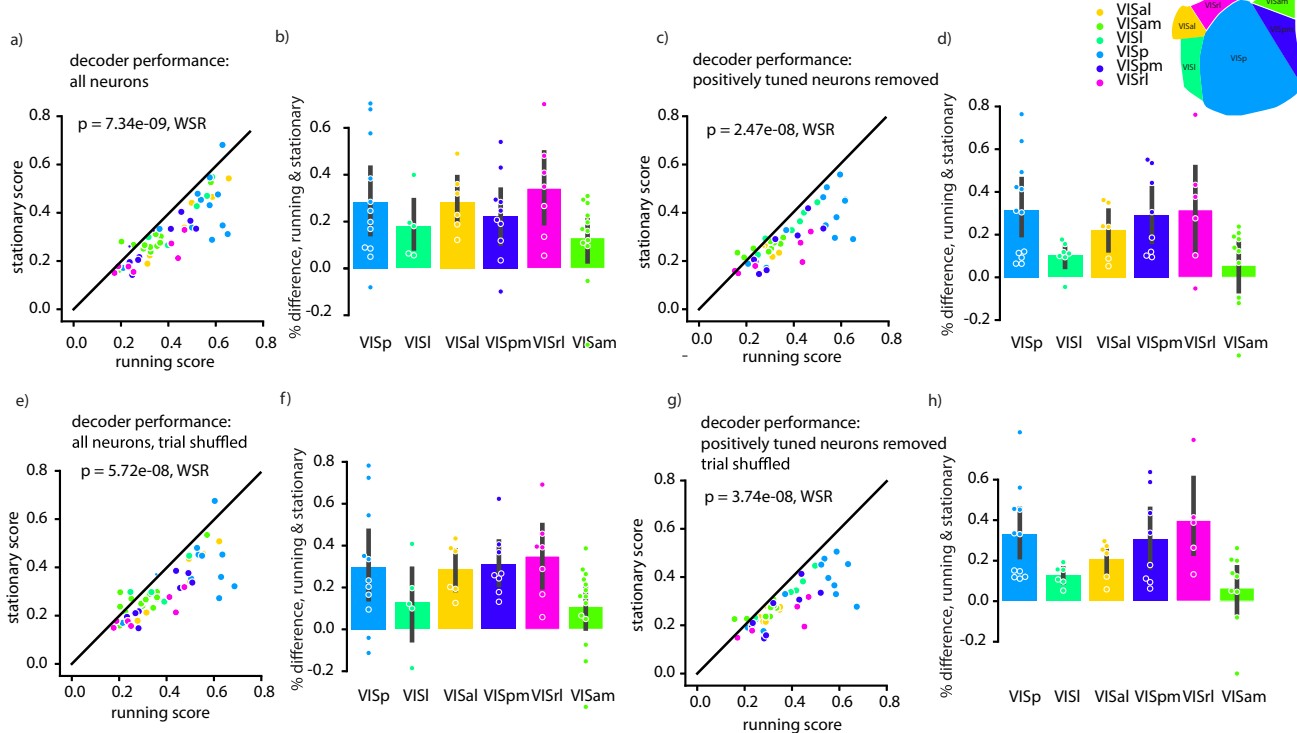

**Fig. 2 Decoding performance (multinomial logistic regression) is improved during running relative to stationary periods, independent of net neural tuning or noise correlations. a** Average fraction of correctly classified visual stimuli during running and stationary periods (average over ten 50:50 train/test splits). Each data point is an individual experiment. Colors indicate brain region recorded. **b** Data from (**a**) displayed separated into visual regions in dataset, only including experiments in which the difference between running and stationary periods was significant (in either direction). Each dot is an individual experiment. **c**, **d** Same as (**a**), (**b**) but excluding neurons that increase their activity during running. **e**, **f** Same as a. but trial-shuffled to remove noise correlations. **g**, **h** Same as a. but excluding neurons that increase their activity during running and trial-shuffled to remove noise correlations. Different decoders, all layers, and differential effect of pupil diameter and running speed are presented in Supplemental Figs. 4–7. Cell and dataset numbers for each region and condition can be found in Supplementary Table 1. All error bars are 95% confidence intervals of mean estimate calculated via 1000 bootstraps. Statistics are calculated via Wilcoxon Sign Rank test, two tailed. Source data are provided as a Source Data file.

the anterior-posterior axis (Fig. 1g). This effect was specific to periods when a stimulus was presented, the sign of the average correlation coefficient was more varied (across all regions) when no stimulus was present (Fig. 1f). These trends were consistent when we separately analyzed data from natural and artificial visual stimuli (Supplementary Fig. 1f, g). Thus, in general, anterior extra striate regions had both a higher fraction of cells tuned to running, and a higher fraction of cells suppressed by running (Fig. 1f, g). When we repeated these analyses using pupil diameter as an indicator of the animals behavioral state instead of locomotion speed, the results were almost identical (Supplementary Fig. 1d), consistent with the interpretation that both locomotion and pupil diameter are indicators of a generally aroused state which underlies the neural changes we observe. This observed net negative correlation of running with neural activity level in anterior extra striate regions is reminiscent of the running/activity correlation previously observed in somatosensory and auditory cortices, and differs from that previously observed in V1. However, all regions we evaluated (including V1) had many cells tuned both positively and negatively to running, leading us to question the relationship between visual encoding accuracy and the overall sign of population-locomotion correlation.

**Population decoding.** To investigate this question, we sought to test whether, despite the diversity of running evoked changes in

magnitude of neural activity, visual coding fidelity (as assessed by decoding performance) was improved in all visual regions[8,13].

To assess decoding performance, we trained a linear classifier (multinomial logistic regression) to decode which of 8 different drifting gratings were presented to a mouse. Decoding performance was significantly higher for responses during running than for responses during stationary periods (Fig. 2a). This trend was also present in each individual visual region (Fig. 2b), despite the fact that in many of these regions, net neural activity decreased during periods of locomotion. Indeed, when we excluded from all datasets any neurons whose activity increased during locomotion, we observed the same improvement of decoding performance (Fig. 2c, d). We also performed a control analysis to try to separate the influence of arousal and locomotion, to determine whether a large variability in behavioral states underlying the "stationary" condition might contribute to poor decoding performance (Supplemental Fig. 6). We did not find an effect of pupil diameter on decoding during the stationary periods. However, it is possible that this is due to the relatively small number of trials included in this analysis, due to constraints of balancing conditions.

To investigate what changes in neural response statistics (other than overall magnitude of response) led to this improvement in classifier performance, we analyzed the noise correlations of population responses. Previous work has shown that noise correlations in V1 decrease during running[7,8], and this has been proposed to be a primary reason for improved decoding accuracy

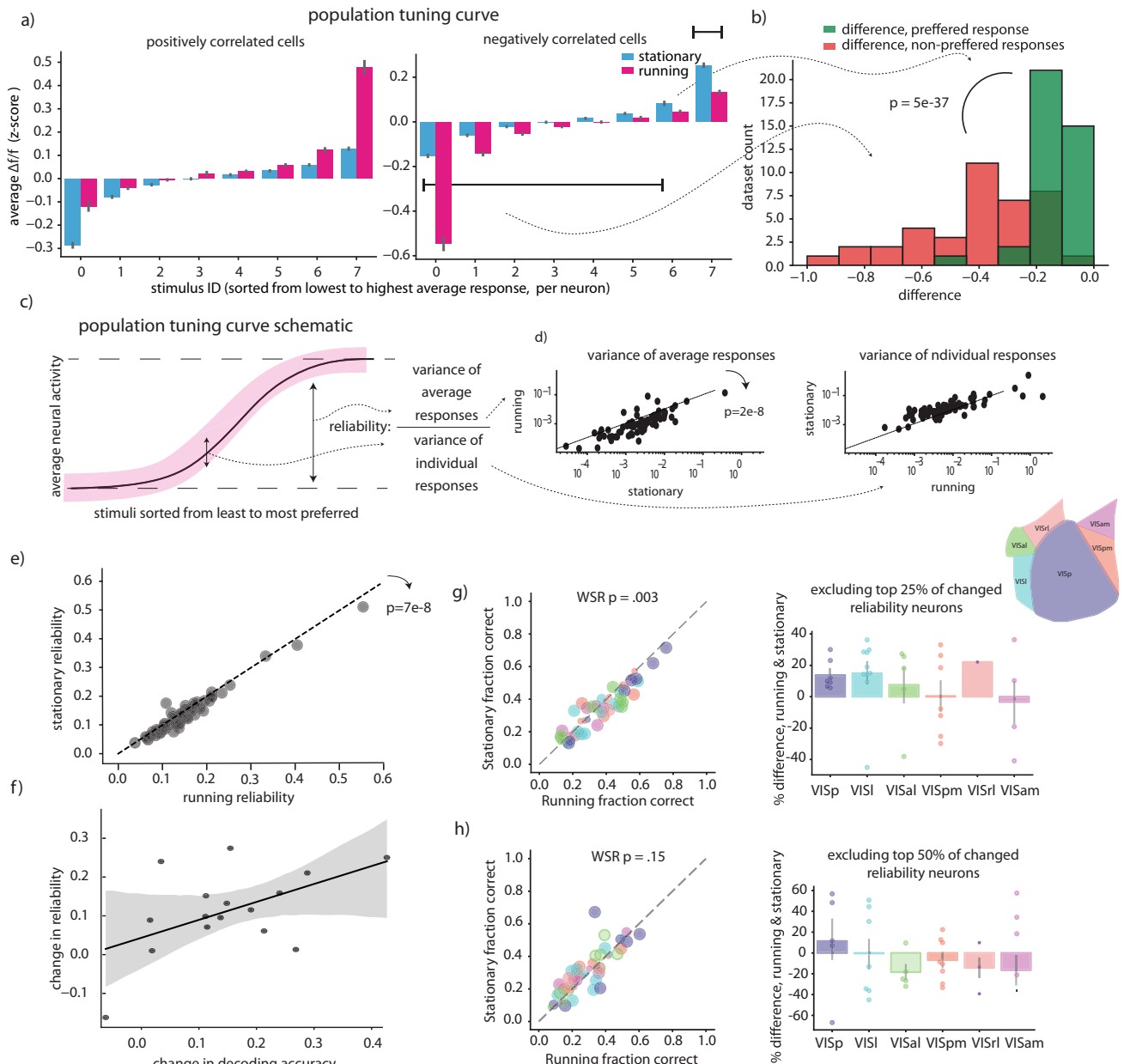

**Fig. 3 Increased reliability accounts for decoding results. a** Population tuning curves separately plotted for neurons that have a negative correlation coefficient with running speed, and those that have a positive correlation tuning curve. Neural responses were z-scored across all stimuli before being split into running and stationary groups, then sorting according to their average response to each stimulus. Dff z-score is averaged across all cells. 2577 cells positively correlated, and 3209 negatively correlated cells were included in this analysis, error bars are 95% confidence on the mean estimate determined by 1000 bootstraps. **b** histogram comparing the average change in neural activity for the preferred stimuli, vs. all other stimuli. Percent difference is averaged across all cells in a particular imaging dataset before plotting. 39 – mouse datasets were included in this analysis, only cells significantly tuned to drifting gratings and running speed were included in this analysis, datasets without enough running or stationary periods to calculate tuning were excluded (see methods). A positive value corresponds to cases where the average response is higher during running, and a negative value corresponds to cases where the average response is lower during running. **c** Schematic of reliability calculation. **d** Scatter plots of variance of average responses and variance of individual responses—the numerator and denominator of the reliability metric, respectively. Values were averaged over all neurons in each mouse- dataset, a total of 61 datasets were included in this analysis. **e** Reliability as defined in panel "**d**", averaged across all neurons per mouse-dataset, in running vs. stationary periods. A total of 61 mouse-datasets were included in this analysis. **f** Correlation between decoding performance and average percent change in reliability in each experiment. Error envelope is defined by 1000 bootstrap resamples and refitting of the linear regression line. **g** Same as 2a, b except decoders trained excluding top 25% of cells with the most changed reliability (**h**). Same as (**g**) except decoders trained excluding top 50% of cells with the most changed reliability. Cell and mouse-dataset numbers as in Fig. 2—Supplementary Table 1. *P* values are Wilcoxon sign rank test against the null hypothesis of equal distribution medians, two tailed. Error bars on histograms are 95% confidence intervals on the mean estimate determined by 1000 bootstraps. ΔF/F (dff): change in fluorescence over baseline fluorescence. See methods for details. Source data are provided as a Source Data file.

during running compared to during stationary periods[13]. Additionally, increased behavioral discrimination performance in mice during cholinergic modulation (which is typically present during locomotion) has been attributed to de-correlated neural activity patterns[3]. To determine whether decreased noise correlations during running epochs were responsible for the increases in classifier performance we observed, we compared decoder performance on data that were trial shuffled. Surprisingly, we found that—although trial shuffling slightly reduced the size of the improvement in decoding accuracy during running—a robust difference in decoder performance between running and stationary periods persisted for shuffled data (Fig. 2e, f). We were nevertheless curious whether a combination of increased response levels during running and reduced noise correlations could account for the difference in decoding performance, as has been previously suggested[13]. We therefore repeated our decoding analysis on a dataset that was trial shuffled after removing all neurons excited by running; surprisingly, the difference in decoding between running and stationary periods was still present (Fig. 2g, h). These trends were consistent across multiple choices of classifier (Supplementary Figs. 4 and 5).

*Neural encoding reliability.* Motivated by these findings, we hypothesized that individual neurons might encode the stimulus identity more reliably when the animal was running, even if their overall activity did not increase. To assess the relationship between stimulus encoding and running-speed correlation, we restricted our correlation analysis to the same drifting gratings stimulus used for our decoding analysis. We first examined the distribution of running speed – activity correlations across the visual regions, similarly to our analysis from Fig. 1. We observed that, although in general drifting gratings responsive neurons tend to be more positively correlated with running than drifting gratings non-responsive neurons (Supplemental Fig. 8a, b), neurons in all regions showed a diversity of running speed – activity correlations. When we examined population tuning curves we found that in cells whose overall activity during the drifting gratings stimulus set increased when running, the primary effect on the population tuning curve was to increase the response to the most preferred stimuli (Fig. 3a, left),. In contrast, in cells whose overall activity during the drifting gratings stimulus set decreased when running, the primary effect on the population tuning curve was a decrease in response to the least preferred stimuli (Fig. 3a, right). Thus, in tuned cells, even a net neural activity decrease during locomotion might increase the signal to noise ratio, and thus the decodability of visual stimuli (Fig. 3b). In these data a selective reduction in background activity (aka activity during the non-preferred stimuli) is the primary cause of the overall activity reduction we previously observed, when marginalizing across all stimuli (Fig. 3d).

Indeed, we found that even neurons whose activity did not increase during running showed increased 'reliability', defined as the variance of each neuron's average response to each image divided by the total variance of each neuron's response[10], and this improved reliability was correlated with the increased decoding performance (Fig. 3e, f). We excluded the top 25% and top 50% of neurons whose reliability changed in each separate experimental session, and retrained and tested decoders. We found that, although the decoders still performed similarly to decoders trained with all neurons, with 25% of neurons excluded the difference between running and stationary decoding accuracy was greatly diminished, and it was abolished when the top 50% of neurons with changed reliability were excluded (Fig. 3g, h). Thus, unlike previous reports based on data from V1, we found increased decoding accuracy correlated with increased fidelity in

single neuron responses and was not explainable entirely by decreased noise correlations or increased overall activity levels.

*Leaky integrate and fire models.* Lastly, we sought to investigate possible physiological mechanisms underlying the increased reliability of single-neuron responses during running. It has been previously reported that during periods of locomotion, background membrane voltage fluctuations of neurons in V1 decrease[8,9]. We performed simulations of leaky integrate and fire neurons (LIF) to determine whether this decreased membrane voltage fluctuation could explain the increased reliability in neurons whose activity either decreased or did not change during running. We added Gaussian noise to the membrane voltage of LIF neurons, while driving them with input current corresponding to randomized presentation of 15 different input current levels, corresponding to Gaussian shaped tuning to the 15 different simulated stimuli (Fig. 4a). As expected, we observed that increasing the peak input current increased the neuron's firing rate and its reliability, and that adding noise to the membrane potential also increased the neuron's firing rate, but reduced its reliability (Fig. 4b, Supplemental Fig. 9c). We chose two noise levels reflective of levels measured in vivo ($19 \ (mV)^2$ and $36 \ (mV)^2$) for running and stationary epochs respectively[4], and simulated responses across a range of peak input current amplitudes (Fig. 4c). We found a sharp increase in reliability of these simulated responses between noise variances of $36 \ (mV)^2$ and $19 \ (mV)^2$, implying that neurons had significant room to decrease or maintain their activity levels while still improving reliability (Fig. 4d). In combination with the observation that lowered background noise itself can lead to lower firing rates without changes the mean synaptic drive to a neuron, our simulations explain how a neuron's activity level could easily be reduced by ~50% during locomotion, while its response reliability nonetheless increased. There are many important features of actual cortical circuits (such as recurrence between visual areas, and even the fact that higher order visual areas receive input from lower order visual regions, which may already have running speed modulated activity), which are not included in these simulations. These simulations are intended to show that all else being equal, simply the changing background noise can cause all of the effects we observe in our data (e.g. would, in the absence of other factors, both decrease firing rates and increase response reliability). Further experimentation and physiological measurements will be required to establish whether a reduction in membrane voltage fluctuations during locomotion explains the enhancement in decoding performance we observed, however our simulations are consistent with this hypothesis.

## Discussion

A normative theory that accounts for the enhancement in both activity and coding accuracy of visual neurons during locomotion proposes that running triggers a visual selective-attention mechanism, as vision is the most navigationally-important sense for rodents[14]. This hypothesis is supported by the observation that correlation to running is largely negative in both motor and somatosensory cortices[15]. However, we observed a striking diversity in the type of neural activity changes during locomotion across different visual cortical regions. Surprisingly, neurons in some higher order visual areas were more likely to be negatively correlated with running than positively correlated with running, in contrast to previous results showing response enhancement V1. This negative correlation is not easily reconcilable with theories that explain the running speed modulation in V1 simply by enhanced 'attention' to vision during running.

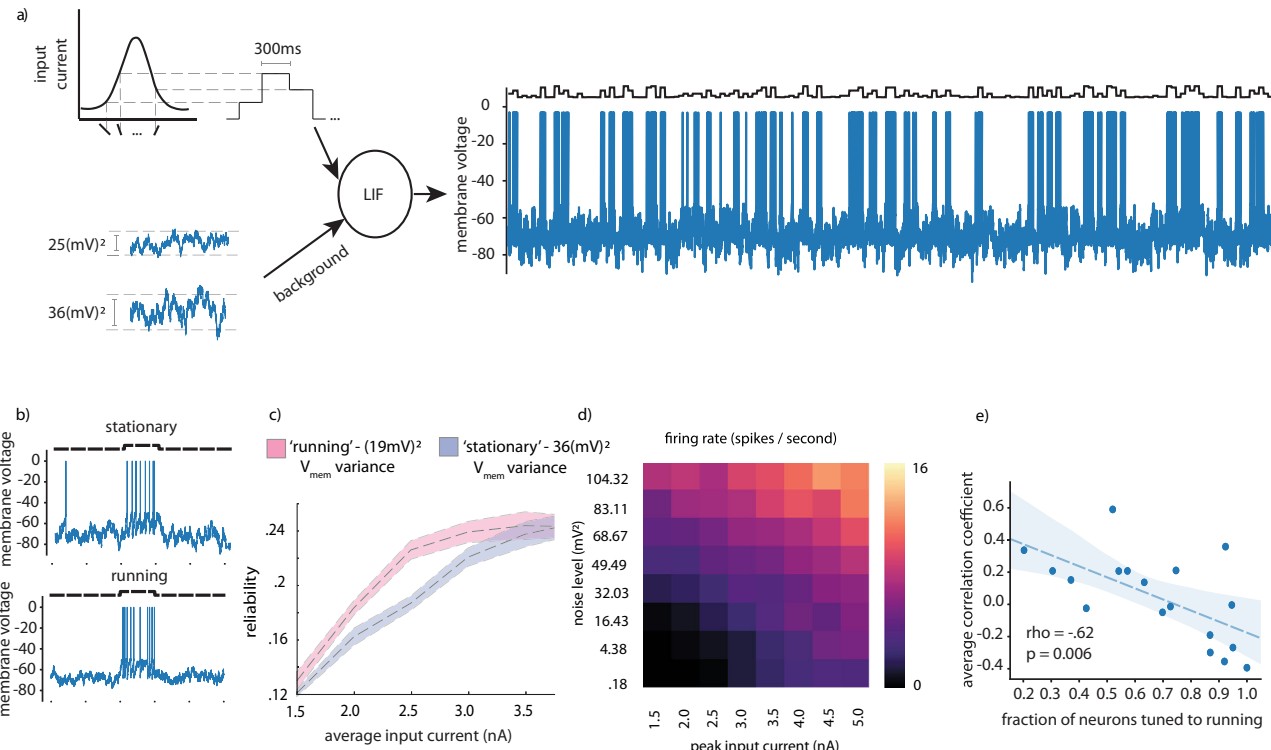

**Fig. 4 LIF model demonstrating how membrane voltage noise can impact neural reliability. a** Schematic explaining LIF simulation with additive Gaussian noise corresponding to membrane voltage measured during stationary (top) and running (bottom) epochs **b** Firing rate versus noise level and peak input current. **c** Example of LIF simulations corresponding to running and stationary periods with current injection. **d** Reliability versus input current for running and stationary membrane voltage noise levels. Error envelope standard deviation over 10 independent runs of model simulation. **e** Correlation between fraction of neurons tuned to running in each region and cre-line and the average correlation coefficient to running measured in that region and cre-line. Wald test with t distribution statistic, against the null hypothesis the slope is zero, two-sided. 18 data points are presented, one for each region and cre-line analyzed. Underlying cell numbers as in Supplementary Fig. 1h, i.

However, despite the trend towards running-induced suppression in higher order visual cortices, running still enhanced the representations of visual stimuli, as measured by decoding accuracy. This effect was not attributable to noise correlations or increased magnitude of neural activity, but instead was explainable by increased reliability of individual neural responses.

Physiologically, both the decrease in activity we observed, and the increase in reliability could result from lower background membrane voltage fluctuations during locomotion[8]. Considering that activity in somatosensory and motor cortex are also negatively correlated with running[15], it is possible that this negative correlation we observe is a more generic cortical phenomenon. However, it is also possible that the running induced negative correlation to running seen in somatosensory cortex and motor cortex[15] are mechanistically distinct from what we see in some higher order visual cortices, as these regions are likely to be spuriously activated by any running related signals, such as interoception, whisking, etc. It's possible the negative correlation to running in motor and somatosensory signals instead acts to dampen these abundant signals.

Alternately, given that the correlation with running speed seems to be dependent on how strongly a neuron is driven by a particular stimulus, it's possible that as a community we are just better at selecting stimuli that specifically drive V1 neurons than almost any other region, and thus V1 is the region in which stimulus-gain effects are the most obvious.

Similarly, our results could be a downstream consequence of a difference in distribution of preferred stimuli type across the difference visual regions—as previous authors have noticed that neurons with particular tuning preferences (namely those tuned to high spatial frequencies[16]) are more likely to be positively correlated with running. This effect has been described as a potential rodent analog of spatial attention. However this is unlikely to completely explain our results–the published data on spatial frequency selectivity in mouse higher order visual cortex[12] does not correlate with the distribution of running speed selectivity we have observed. For example, in Marshel et al. AL has the lowest spatial selectivity, and PM has the highest spatial selectivity, whereas in our data AL and PM have relatively similar overall distributions of neural correlations with locomotion speed.

An important caveat to our work is that although we have only measured and analyzed Ca2+ fluorescence as a proxy for neural activity, many previous theoretical studies on neural tuning, neural response gain, and even our own work on neural reliability through LIF simulations relied on analyzing neural firing rates. Ca2+ activity is a fundamentally different measure of neural activity and in many cases only bears a noisy, non-linear relationship to neural firing rates. In particular Ca2+ sensors display a non-linear activity/fluorescence relationship, with saturation both at high and low firing rates, which could potentially impact analysis of neural response reliability, and overall correlation to running. In particular, small changes in neural activity when firing rates are low can be difficult to detect with GCaMP. This could limit our ability to detect decreases in spontaneous activity and decreases in activity evoked in response to presentation of non-preferred stimuli. Further work should be done to verify these findings with electrophysiology.

In either case, these results highlight previously unknown differences in running speed modulation in primary visual cortex and posterior and anterior higher order visual cortices and highlight important questions for future work.

## Methods

**Data collection**. We analyzed data from the publicly available Allen Institute for Brain Science Brain Observatory data set. Their full data collection methodology can be found in the white paper[11]. In brief, transgenic mice expressing GCaMP6f in laminar-specific subsets of cortical pyramidal neurons underwent intrinsic signal imaging to map their visual cortical regions before cranial windows were implanted above the desired visual region. Mice were habituated to head fixation before imaging sessions in which they were shown either a mean luminance blank screen, natural scenes, natural movies, locally sparse noise, or drifting or static gratings. These data were collected over four different experimental session per mouse, each of which lasted ~1 h. Stimulus types were interleaved throughout each session, for example in session A mice are presented with 10 min of drifting gratings, followed by 10 min of a natural movie, followed by 5 min of a different natural movie, followed by another 10 min of drifting gratings, followed by 5 minutes of spontaneous activity (mean luminance blank screen), etc. The full details of each stimulus design, and the order in which they were presented is described in the Stimulus Set and Response analysis white paper in the Allen Institute documentation (http://help.brain-map.org/display/observatory/Documentation?preview=/10616846/10813485/VisualCoding_VisualStimuli.pdf).

Neuropil corrected normalized fluorescence change (ΔF/F) traces for each cell were extracted using automated, structural ROI based-methods (See Allen Institute white paper for details). ΔF/F was calculated using a rolling-window baseline method. The baseline was calculated as a windowed mean of the windowed mode. The mode kernel width is 5400 frames (3 min), and the mean kernel width is 3000 frames (1.667 min). The raw, neuropil-corrected fluorescence was then baseline-subtracted and baseline-normalized. Eye movements, pupil area and locomotion speed were recorded for some experiments, here we only analyzed experiments for which pupil area was available. We analyzed data from the Cux2-Cre-ERT2 (layer 2/3), rbp4-Cre (layer 5), and Rorb-Ires-Cre (layer 4) mice, as data from these transgenic lines were available for all regions in the data set.

**Running speed tuning**. For our analyses of running speed tuning, we selected mice who ran for at least one quarter of the stimulus presentation period (to ensure enough data-points to accurately calculate tuning). We also excluded mice whose maximum running speed was less than 15 cm per second, to ensure enough of a range of running speeds were present to accurately assess correlations. We estimated running-speed tuning curves by first binning data into 20 equal-sized bins (i.e., 20 quantiles) of running speed, ranging from zero to the maximum speed attained by each mouse, and taking the mean neural activity in each bin, effectively creating a non-parametric 'tuning curve' to running. To determine whether a neuron's activity was significantly modulated by running, we compared the neuron's running-speed tuning curve to a running-speed tuning curve computed from randomly permuted data using Levene's $t$ test of variance; we considered a neuron tuned if its non-shuffled tuning curve had significantly more variance than it's shuffled tuning curve[5]. We calculated Spearman's rho on the binned data, to determine whether each neuron was monotonically tuned to running. We considered neuron with a statistically significantly negative rho to be negatively correlated with running, and a statistically significantly positive rho to be positively correlated with running. Neurons whose activity were not statistically significant monotonically correlated to running speed, but that had passed levene's $t$ test of variance were considered "non-monotonically" tuned to running. We used an alpha-level of $p = 0.05$ for our estimate of significant tuning. We calculated this tuning both separately for natural (natural scenes and natural movies), artificial (drifting gratings, static gratings, and noise stimuli), and spontaneous activity. We found that tuning was similar across natural and artificial stimulus conditions, and therefore grouped them together in the main text (but see Supplemental Fig. 1). Note, during the presentation of some stimuli there were blank periods in between subsequent stimuli presentations, e.g. between presentation of different orientations of drifting gratings. These data were discarded for this analysis, and were not considered as part of either the spontaneous or stimulus datasets. When we refer to "spontaneous" periods, we are referring to the 10 min blocks of mean luminance blank screens during each experimental session (a total of 30 minutes of data for each mouse).

**Gaussian tuning model**. As in Saleem et al.[5] three different Gaussian models were fit to the neural tuning curves. The models were differentiated by constraints on the locations of the center parameter. The first model was a monotonically increasing model where we constrained the center of the Gaussian to fall higher than the highest running speed, the second model was a monotonically decreasing model where we constrained the center of the Gaussian to fall lower than the lowest running speed, and the third model was one where the center was allowed to fall within the highest and lowest values. We determined which model fit the data best 10 repetitions of a cross validation procedure where we split the data into 75% training, 25% test set before creating the tuning curves that we fit the models to.

**Drifting gratings analysis**. Drifting grating selectivity indices were obtained via the AllenSDK. Again, the full details of the calculations performed by the Allen institute can be found here: http://help.brain-map.org/display/observatory/Documentation?preview=/10616846/10813485/VisualCoding_VisualStimuli.pdf.

Neurons were classified as significantly responsive to drifting gratings if a one-way anova between all responses obtained during all the different stimulus conditions (different grating direction and temporal frequencies) was less than 0.05. Drifting grating selectivity index was defined as ($R_{preferred} − R_{null}$) / ($R_{preferred} + R_{null}$) where $R_{preferred}$ is the average neural response to the stimulus that neuron responded the most to, and $R_{null}$ is the average neural response to the stimulus orthogonal to the preferred stimulus.

**Decoding analysis**. We performed all decoding analyses on the Drifting Gratings dataset from the Allen Institute Brain Observatory. Details of this dataset can be found in their white paper[11], but briefly: each mouse was presented with 75 repetitions of 8 drifting gratings of different directions, for 2 s per presentation, with a 1 s blank period in between stimuli. Each grating presentation had a spatial frequency of 0.04 cpd and a temporal frequency randomly selected from a set of 5 different temporal frequencies. We performed decoding of grating direction while ignoring temporal frequency. For the purposes of the decoding analysis, we excluded individual experiments in which fewer than 10 neurons were recorded—this exclusion criteria mainly applied to lower levels of VISrl in the analyses when considering all neurons except those with positive tuning to running. Note that decoding analyses included data from additional mice that were excluded from the speed tuning curve analyses (due to insufficient time spent running). Decoding analyses that excluded neurons with positive tuning to running, however, only included data from the subset of mice whose tuning curves had been well characterized.

To perform decoding, we extracted a vector of neural population activity for each trial by normalized averaging fluorescence (ΔF/F) over a 2 s window that was offset by 330 ms (10 imaging frames) from the beginning of stimulus presentation. This time window was chosen by selecting the window that maximized the $R^2$ prediction performance of held out trials from the PSTH. To compare decoding during running vs. during stationary periods, we split the data into "running trials" (trials with average velocity > 3 cm/s, but whose minimum velocity did not drop below 0.5 cm/s) and "stationary trials" (trials with average speed < 0.5 cm/s but whose maximum speed did not exceed 3 cm/s). We randomly sub-sampled the data to ensure equal number of trials per visual stimulus class in both running and non-running subsets. For the data presented in the main paper, we performed decoding of neural responses using an 8-way multinomial logistic regression (MLR) classifier, as implemented in the scikit-learn[17] python package. Classifier weights were learned via the LBFGS algorithm, and no regularization was applied. Classifier performance was assessed via a cross-validation procedure: fraction of correctly labeled stimuli on a test set comprising 50% of the data was averaged over 10 random (class balanced) train-test splits. For shuffling analyses, we randomly permuted each neuron's responses across trials (within the same class) so that population response vectors contained non-simultaneous responses, breaking trial to trial correlations between simultaneously recorded neurons. In the figures, decoders were both trained and tested on shuffled data, although in separate analysis we either only tested or only trained on the shuffled data, without significantly different results. The main results are consistent across different decoder types, with results obtained with Gaussian Naïve Bayes and Multi-Class Support Vector machines (evaluated in a one-vs-rest scheme) are presented in the Supplemental Materials.

We assessed each cell's reliability, defined as the variance of the average ΔF/F response across different stimuli divided by the total variance across responses to all stimuli[10]. Thus, if a cell has high variance of response across all stimuli (e.g. a greater dynamic range in its responses to different stimuli types), with low trial to trial noise, it is considered to be extremely reliable – thus, a single measurement of that neuron's response contains a large amount of information about stimulus identity.

**Leaky integrate and fire simulations**. To examine the possible effects of membrane voltage fluctuation on response reliability, we performed simulations of leaky integrate and fire neurons, using Euler's method. Membrane voltage was initialized to 70 mV, and then integrated over time according to the following differential equation:

$$C_m \left( \frac{dV}{dt} \right) + g_l (V - E_l) = I_{app} \tag{1}$$

$$\text{If } V > V_{thresh}, \ V = E_l \tag{2}$$

$C_m$ : *membrane capacitance*

$g_l$ : *leak conductance*

$E_l$ : *resting membrane potential*

$I_{app}$ : *applied current*

$V_{thresh}$ : *Threshold voltage*

The following parameters were used Vthresh: −49 mV, Vinit: −70 mV, integration time step = 0.05 ms, Cm = 490 pF, gl = 16 pS, El = −65 mV. For Supplementary Fig. 7, we did an additional simulation with resting membrane potential (El) of 68 mV and 70 mV for running and stationary periods respectively, to match the resting membrane potentials measured in vivo[4],[9].

We created a Gaussian shaped tuning curve across 15 different stimuli to define the input current generated by each stimulus (as illustrated in Fig. 4). We simulated different levels of membrane voltage fluctuation by adding independent Gaussian noise to membrane voltage at each time step. Noise variance levels of 19 mV$^2$ and 36 mV$^2$ were empirically determined to match values recorded in vivo for running and stationary animals. We simulated presenting 300 ms trials of each stimulus 10 times in a randomized order, and calculated reliability in the same fashion as described above.

**Reporting summary**. Further information on research design is available in the Nature Research Reporting Summary linked to this article.

## Data availability

Data is freely available through the Allen Institute Brain Observatory[11]. Source data and minimal plotting code to recreate the figures are included with this paper.

## Code availability

Code to reproduce all figures is freely available on the authors github repository github.com/achristensen56/AIBSmouselocomotion.

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

## Acknowledgements

A.J.C. was supported by a Texas Instruments Stanford Graduate Fellowship; J.W.P. was supported by grants from the McKnight Foundation, Simons Collaboration on the Global Brain (SCGB AWD1004351) and the NSF CAREER Award (IIS-1150186).

## Author contributions

A.J.C. and J.W.P. conceived the study, A.J.C. performed the analyses, and A.J.C. and J.W.P. wrote the manuscript.

## Competing interests

The authors declare no competing interests.
