## [Peer Review File · Nature Communications]

Reduced neural activity but improved coding in rodent higher-order visual cortex during locomotionReviewers' comments:

Reviewer #1 (Remarks to the Author):

Christensen and Pillow investigated how locomotion impacts neural activity in higher-order visual cortex in mice, using large datasets from Allen Brain Institute. They report that the neural activity is suppressed during locomotion in the higher-order visual cortex, while the activity is enhanced in the primary visual cortex (V1). The authors further show that, despite this suppression, visual activity could be decoded more accurately during locomotion. The improvement in the decoding accuracy was due to higher reliability of visual responses in individual neurons. The higher reliability can be modeled by leaky integrate and fire neurons with decreased membrane potential variance.

The main finding of this study, reduced neural activity in higher-order visual cortex during locomotion together with the improved decoding capability, is a compact and interesting finding that might be suited for a brief report. However, the manuscript also needs a number of improvements.

1) The definition of df/f

For the calculation of df/f , the manuscript refers to Allen Institute, "See Allen Institute white paper for details". According to Allen Institute website, "For each trial, the $\Delta F/F$ for each cell was calculated using the mean fluorescence of the preceding 1 second as the baseline F_0 (Fig. 5A).

$$DF/F = (F - F_0)/F_0$$

([http://help.brain-](http://help.brain-map.org/display/observatory/Documentation?preview=/10616846/10813485/VisualCoding_VisualStimuli.pdf)

[map.org/display/observatory/Documentation?preview=/10616846/10813485/VisualCoding_VisualStimuli.pdf](http://help.brain-map.org/display/observatory/Documentation?preview=/10616846/10813485/VisualCoding_VisualStimuli.pdf))

This raises multiple concerns.

a) With this definition, F_0 will be affected by the running speed of the mice during the baseline period. Thus, df/f during visual stimulation will also be affected by the running speed during the baseline period, which makes the interpretation of df/f complicated. My suggestion is to calculate F_0 using a sliding window of a few minutes (e.g., Peron et al., 2015) to make it more robust against rapid changes in the running speed.

b) df/f signal in GCaMP6f is not equal to firing rate: GCaMP6f has a rise time of 45ms and a decay time constant > 100 ms (e.g., Supplementary Table 3 of Chen et al., 2013). Is the decay of the fluorescence signals considered in the analysis of this study, for example when binning the data based on the running period? This issue will be particularly relevant when the mice change their running speed during the baseline or visual stimulations.

c) The decoding analysis is based on the averaged df/f over 2s. It is unclear how the averaged df/f is calculated when the mice changed their running speeds during the stimulus presentation (or during the 1s baseline period). Are these trials discarded, and if so what are the proportions of the trials that are discarded in each mouse?

d) How was df/f defined when no stimuli were presented (e.g., "spontaneous" in Figure 1f and Supplementary Figure 1g)? If the baseline F_0 is the mean fluorescence of 1 second period preceding the no stimulus presentation, df/f following this baseline period would be around 0, regardless of the running speed.

These pieces of information need to be described in much more detail in Methods. In addition, example traces of df/f would be helpful for the readers to capture what the data quality is and how the analysis was processed.

2) The relationship between visual responses and locomotion tuning

Despite that this study is on the neural coding in the visual cortex, there are no attempts to investigate the relationship between visual-response tuning vs. locomotion tuning. Do visual responses and the effect of locomotion add up linearly? Supplementary Figure 1e-g suggests that the effect of locomotion on spontaneous activity could be different from that on visual responses at

the population level. This could imply complex effects of locomotion on visual responses. More detailed descriptions on the non-linear effects are necessary.

3) Tuning to the running speed

The authors showed example cells that are tuned to specific running speed at the top of Figure 1b, and yet, used Spearman's rank correlation coefficient to capture the population trend. The correlation coefficient of the cells like those at the top of Figure 1b can show strange values. It is necessary to distinguish the cells that are monotonically increasing/decreasing and the cells that are tuned to specific running speeds more carefully.

4) Leaky integrate and fire model

a) The Methods do not contain essential information related to the model. For instance, the equation is missing, and it is not clear how the "input current drawn from Gaussian shaped tuning to 15 different objects" are related to the current range described in the manuscript

b) The model assumes that the visually evoked currents are exactly the same between stationary and running periods, but this assumption would not hold for the neurons in higher-order visual cortex, as they receive inputs from neurons in V1 (that has higher reliability during locomotion).

c) Does this model assume that the baseline membrane potential is the same during running and stationary periods? Is this assumption consistent with the data in Polack and Golshani, 2013 (from which the parameters of the model derive)?

5) Relationship between pupil size and neural activity

One of the popular ways to assay the "arousal" (page 2) level is to measure the pupil diameter, which has also been employed in the study of the primary visual cortex (e.g., Vinck et al., 2015; Reimer et al., 2014). Since "Eye movements and locomotion speed were recorded" (Methods), I would expect analysis on the relationship between the pupil diameter and visual responses included in this manuscript.

6) Some of the descriptions in the introduction and the discussion need improvement.

a) In page 1, "However, the investigation of these effects has so far been limited to primary visual cortex (V1)." This statement is wrong. The effect of locomotion on visual responses was investigated in areas AL and PM in a pioneering work by Andermann et al., 2011. In fact, Andermann et al reported no significant suppression in PM, which contrasts with the current study. This issue needs to be discussed in detail. To find out the discrepancy, it is advisable to analyze the Allen database in the same way as Andermann et al.

b) It is not clear how the second hypothesis described in the Introduction can be addressed by Allen's datasets. Could this be clarified?

Minor:

1) Please include the number of cells in the figure legends.

2) "average fluorescence (df/f)" (for example Figure 1b)

df/f is not fluorescence, but an index normalized to the fluorescence at the resting state to capture the neural activity.

3) P3: Figure 1c. "dff" should be "df/f".

4) P4: "These trends were consistent across multiple choices of classifiers (Supplemental figure 2)" Supplemental Figure 2 shows the results for one classifier. Did the authors test only two classifiers in total?

5) P4: "reduced noise curious could" "curious" should be "correlation"

6) P5: (figure 2k). This should be figure 3d.

7) P6: mV^2 should be represented as $(mV)^2$

8) P6: (figure 3k). This should be figure 3e-g.

Reviewer #2 (Remarks to the Author):

This manuscript, by Chistensen and Pillow, represents a very new kind of neuroscience. In this case, the authors have sourced a primary data set from the Allen Institute Brain Observatory, a repository of 2-photon imaging data collected in the visual areas of awake mice. Using these data, they have investigated the relationship between locomotion and firing rates in primary and higher-order visual cortices. The question of state-dependent regulation of neural activity and gain control is a central one in the field, and could potentially be very interesting as a starting point for examining the flow of information among higher-order cortical areas. However, the authors have not addressed this issue in a manner leading to a clear set of answers. Indeed, it is uncertain that the dataset they have chosen would allow any such clarity, due to inherent restrictions. Furthermore, the nature of the dataset precludes addressing these problems by changing the experimental paradigm, and I do not see any way for the manuscript to be substantially improved under these conditions.

Overall, the authors seem to be mistaking 'modulation of firing rate by running speed' for 'locomotion-induced gain modulation of visual responses'. Firing rate alone is not interesting, and the experimental paradigm employed here is not optimized to answer the main questions posed by the authors. My main concerns are outlined in more detail below.

First, because there are still few or no papers published from the Allen data set, the onus is on the authors to perform appropriate due diligence. It is not sufficient to reference the Allen white paper. The authors need to provide some quantitative assessment of the quality of the data. In fact, the authors explicitly state that they did no post-processing of the data once downloaded, suggesting that ROIs were not examined for overall activity, stability of activity over time, or visual responsiveness. Furthermore, the authors present neither traces from individual ROIs nor statistics on ROI quality or criteria for inclusions—all elements that would allow the reader to evaluate the quality of the data set.

A number of key points about the analyses and the data are missing from the manuscript, leaving the interpretation of the data very unclear. What method for dF/F was used in the final analysis? How was the baseline calculated? Several methods for baseline estimates have been used consistently in the field, including normalizing to the mean across an imaging session and normalizing to the lowest 10% of the values. However, these methods affect the range of values expected for spontaneous and evoked activity. Along similar lines, what percentage of cells were deemed visually responsive, according to what criteria?

Presumably the mice are running in lighted conditions, since visual stimuli are shown, but this is not stated. Using data from light-only periods restricts the authors' ability to compare their findings with other groups', since recent work has distinguished locomotor responses in local V1 circuits occurring during dark and light running periods. Comparing locomotion-associated gain changes in dark and light conditions would enhance the authors' ability to draw comparisons between V1 and higher-order areas.

It is unclear which periods the 'spontaneous' data are taken from. Because the Allen white paper suggests continuous visual stimuli, 2s on and 2 s off, the 'spontaneous' data are presumably taken from the inter-stimulus intervals. Even for gCAMP6f, which has relatively rapid kinetics, this means that several hundred milliseconds of the 'spontaneous' intervals between visual stimuli are in fact contaminated with the offset tail of the calcium signal. This makes the comparison of the 'visual' and 'spontaneous' datasets problematic. None of the necessary controls or data are presented to address this issue. The authors should instead present stimuli with > 1s between stimuli to reduce the impact of this issue.

The authors suggest that the main point of the study is to examine the 'prevalent view that running acts to enhance visual representations,' but none of the analyses directly address visual representations at all. The issue of locomotion-induced changes in visual gain is quite complex, because the locomotion-associated increase in firing rate observed in V1 and the change in visual SNR are potentially mechanistically distinct. Here the authors have calculated only dF/F measurements, as a proxy for firing rate. Firing rate is not sufficient to address modulation of visual gain and encoding, and the authors should instead calculate a measure of SNR or visual response amplitude. In addition, the analysis of visual responsiveness should be restricted to cells that show a significant modulation by visual stimuli under at least one condition—up to 25% of V1 cells may not respond to visual stimuli under these experimental conditions.

A related issue is that the spontaneous and visual datasets are conflated, and the visual stimulus properties are constantly varying. This presents a challenge for calculating SNR. A better experimental paradigm would be to present fewer, or one, stimuli many times in succession to examine visual response modulation by running. Alternately, the authors might concentrate on orientation tuning or contrast gain.

Reviewer #3 (Remarks to the Author):

In this manuscript, the authors characterize how locomotion affects neural activity in different areas of mouse visual cortex. They find running-related fluorescence increases in V1 and two other areas, and decreases in AM. Two other areas (PM, RL) have smaller decreases. They use decoding approaches to study which factors (amplitude, correlation, reliability) control changes in information about visual stimuli between running and stationary conditions. The paper is well-written and clear.

The question of how running changes information content is of general interest. However, there are several concerns that cast doubt on the findings as stated.

Major:

1. The abstract says "... suppressed firing in higher-order visual areas. Despite this reduction in gain, visual responses during running could be decoded more accurately than ... during stationary periods." Figure 1c seems to show that LM (VISI), and AL increase fluorescence, contrary to the abstract claim, and only AM shows a substantial decrease during running. And Fig. 2b does not provide clear evidence that AM (pink/magenta) shows an improvement in decoding performance during running. The work seems to support instead the idea that V1 increases firing rate with running, that V1 carries the largest amount of information about the stimulus, and that V1 information is increased the most by running (Fig. 2). Perhaps additional analyses, focusing on AM and examining visual gain in more detail, could provide support for the present claim, but it seems that a major rewrite would be required.

2. The authors use the word "firing" in the abstract, but relating calcium responses to firing can be difficult. The conclusion about reliability -- that increased df/f reliability implies increased spiking reliability -- is especially fraught. Changes in intracellular calcium concentrations and/or buffering could underlie changes (as just one example, perhaps ACh input changes calcium concentrations during running). GCaMP could also cause increased reliability, for example if the fluorescence response is moved closer to saturation by an experimental condition. It might be possible for the authors to support the claim that fluorescence reliability change is due to spike reliability change, but it seems difficult.

Minor:

- It was difficult to determine whether changes in response were calculated during spontaneous periods, or in response to visual stimuli (as is most appropriate when "gain" is used). The df/f changes from Supp. Fig. 1 should be in the main text. And do results change if exclusion of excited neurons is based on artificial/natural/spontaneous responses?
- Response gain should be shown by plotting response to the same visual stimuli in running and stationary states both at the single-neuron and population level.
- The supplementary materials should include the equations of the LIF model and the manuscript should justify leaving out recurrent responses to input, which seem likely to influence spike responses.

Reviewer #4 (Remarks to the Author):

The authors explored the Allen Brain Institute database to discover that, during locomotion, the firing rate in V1 increases (consistent with previous reports) but, surprisingly, it suppresses responses in higher visual areas.

Despite this decrease in firing rate, the reliability of the responses increased, allowing a more accurate decoding of the visual stimulus.

Moreover, the authors show that a simple, leaky integrate and fire neurons may replicate the observed phenomenon if reductions in noise are allowed to counteract the decrease in firing rate during locomotion.

Major comments

One major concern is that most analyses are performed at the population level, in contrast with past studies that investigated how the tuning curves of individual neurons change between rest and locomotion. These data appear available from the Allen dataset. The study could be improved substantially by describing and modeling the changes in tuning across each of the areas to better understand how changes in gain and mean response varies as a function of behavioral state.

There seems to be an interchangeable use of mean spike rates and gain in the manuscript that is somewhat confusing. For example, the authors write "Nevertheless, the observed suppressive effects of running on firing rates contradicts the naïve hypothesis that running induces selective attention to vision that increases the gain of responses throughout visual cortex". However, the two are different concepts and mean spike rates and gain of responses may go in opposite directions (see discussion in Mineault et al, J. Neurosci, 2016). Changes in gain could be evaluated if the authors analyzed changes in the tuning curves of neurons during rest and locomotion, as suggested above. It is not clear how the estimated changes in gain from the present analysis. From the text, it appears they believe changes in gain and mean responses refer to the same phenomenon, but that is not how the terms are used in the literature.

The authors mention that decreases in noise in a simple LIF neuron could potentially replicate some of the phenomena observed, but they do not explain how this could happen. Given that V1 exhibits higher firing rates and increased reliability during locomotion, one would expect the input signals to higher visual areas to reflect this as well. Perhaps, the increased input rates engages local inhibition in a way that the overall mean rate in higher visual areas decreases, despite an increase in input. Once again, a cell-by-cell analysis of how tuning curves change between rest and locomotion may shed light into what is actually happening.

Overall, this reviewer felt the reported phenomena to be interesting, but that more analyses on at the single-cell level are required to better understand how locomotion is changing the activity of neurons. Given that the data are available, it may not be very difficult for the authors to carry such analyses.

Minor comments

Fig 1. A population analysis of the tuning curves would be helpful. Are tuning curves about the same in all visual areas? Are some types (monotonically increasing/decreasing) predominantly found in some areas?

How was Fig 1d put together? Is this a single mouse?

Fig 2. How is decoding performance defined? I could not find this information in the "Decoding analysis" section. It would help to explain what is being plotted in the Fig legend for easy access.

Fig 3. It would help to add a plot of reliability against noise level and mean response (instead of just the peak current). Mean firing rate will increase with both noise and peak current. This will help the reader evaluate better the claim that "that neurons had significant room to decrease their firing rates while still improving reliability."

Typo: "and reduced noise curious could" Remove "curious"

It would be useful to cite a relevant reference, Mineault et al, Enhanced Spatial Resolution... J Neurosci, 2016, which provides a framework to discuss how changes in gain and mean rate may relate to each other.

We do not yet know how attention modulates activity across different visual areas. Thus, it is not clear that the findings rule out increased changes in attention, but simply show that increased decoding accuracy does not necessarily need to be accompanied by increases in mean firing rate.

What were the distributions of spontaneous activity across all these areas? Is there a correlation between the spontaneous activity and the mean changes observed during locomotion across areas? Is there a correlation even within an area (perhaps cells with low spontaneous rate show increases while cells with high spontaneous show decreases)?

We thank the reviewers for their careful reading of our paper, and for their thoughtful comments and insightful suggestions for improving it. We genuinely enjoyed incorporating the reviewer's comments, and we feel they have helped us to significantly improve our manuscript, which we have now renamed: "**Reduced neural activity but improved coding in rodent higher-order visual cortex during locomotion**". Below we provide detailed line-by-line responses to reviewer comments, but first we describe several major changes that were motivated by comments from multiple reviewers:

- 1. New analyses of interaction between running and visual responses (new Fig 2):** all reviewers commented that our previous analysis did not adequately address the issue of locomotion related modulation of **specific** visual stimulus responses, and that this was a topic they were eager for us to address. We examined the relationship between neural activity and locomotion during presentations of drifting grating stimuli. We found that negative correlations between running and firing rate were predominantly due to reduced responses to anti-preferred stimuli; responses to preferred stimuli were relatively unchanged. In other words, the animal's locomotive state seems to change background neural activity levels (if we assume that responses to anti-preferred stimuli are essentially background activity levels), but not activity levels in response to preferred stimuli. Especially when locomotion is correlated with lower neural activity levels (as we report is the dominant running speed tuning type in the anterior higher-order visual regions PM, AM, and RL) this may actually lead to higher reliability, despite lower overall firing rates. This new data fits in very well with the simulation experiments we had already performed for figure 4 – in fact the simulations suggested such a regime could exist!
- 2. New analyses of decoding reliability (panels added to Fig 4):** we performed new analyses and added a set of figure panels (4a-d) that solidified our hypothesis that the change in decoding performance was attributable to improved reliability in single neurons. We calculated the change in reliability in each individual neuron, and then removed from each decoding experiment the top 25 or 50 percent of neurons whose reliability changed between decoding running and stationary periods. These manipulations significantly reduced (or abolished, respectively) the difference between decoding performance in running and stationary periods.
- 3. Improved precision of wording.** Multiple reviewers (rightly) commented that our language choices were sometimes imprecise and did not reflect what the data actually showed. To address this, we have used the phrase "activity is negatively correlated with running" instead of saying "reduced firing" or "suppressed firing". This is both to respect the difference between calcium fluorescence and firing rates, and to be more precise about exactly what it is we are reporting (a correlation with running as opposed

to running induced gain or suppression). We have also removed usage of the term “gain”, and instead refer to “increased activity” (or “decreased activity”).

Below, we provide detailed responses to individual reviewer’s comments in turn.

Reviewer #1:

Christensen and Pillow investigated how locomotion impacts neural activity in higher-order visual cortex in mice, using large datasets from Allen Brain Institute. They report that the neural activity is suppressed during locomotion in the higher-order visual cortex, while the activity is enhanced in the primary visual cortex (V1). The authors further show that, despite this suppression, visual activity could be decoded more accurately during locomotion. The improvement in the decoding accuracy was due to higher reliability of visual responses in individual neurons. The higher reliability can be modeled by leaky integrate and fire neurons with decreased membrane potential variance.

The main finding of this study, reduced neural activity in higher-order visual cortex during locomotion together with the improved decoding capability, is a compact and interesting finding that might be suited for a brief report. However, the manuscript also needs a number of improvements.

1) The definition of df/f

For the calculation of df/f, the manuscript refers to Allen Institute, “See Allen Institute white paper for details”. According to the Allen Institute website, “For each trial, the $\Delta F/F$ for each cell was calculated using the mean fluorescence of the preceding 1 second as the baseline F_0 (Fig. 5A). $DF/F = (F - F_0)/F_0$ ”

(http://help.brain-map.org/display/observatory/Documentation?preview=/10616846/10813485/VisualCoding_VisualStimuli.pdf)

This raises multiple concerns.

a) With this definition, F_0 will be affected by the running speed of the mice during the baseline period. Thus, df/f during visual stimulation will also be affected by the running speed during the baseline period, which makes the interpretation of df/f complicated. My suggestion is to calculate F_0 using a sliding window of a few minutes (e.g., Peron et al., 2015) to make it more robust against rapid changes in the running speed.

Our apologies for the confusion, we should have more clearly specified the preprocessing steps used to generate the data we analyzed. There are two different quantities the Allen Institute refers to as $\Delta F/F$, one is a stimulus response $\Delta F/F$, the other is a stimulus agnostic $\Delta F/F$. The stimulus agnostic $\Delta F/F$ is calculated from the raw calcium fluorescence traces, regardless of any trial structure of the stimuli, using a rolling baseline as the reviewer suggests. The baseline is calculated over a period of 3 minutes. It is this $\Delta F/F$ value we are referring to throughout the manuscript and in the methods section, we do not use their stimulus response $\Delta F/F$ (e.g. where the baseline is the blank period preceding the stimuli). We have added details regarding the calculation of $\Delta F/F$ to the methods of the paper (quoted below), and hope this is now clear!

“... $\Delta F/F$ was calculated using a rolling-window baseline method. The baseline was calculated as a windowed mean of the windowed mode. The mode kernel width is 5400 frames (3 minutes), and the mean kernel width is 3000 frames (1.667 minutes). The raw, neuropil-corrected fluorescence was then baseline-subtracted and baseline-normalized...”

b) df/f signal in GCaMP6f is not equal to firing rate: GCaMP6f has a rise time of 45ms and a decay time constant > 100 ms (e.g., Supplementary Table 3 of Chen et al., 2013). Is the decay of the fluorescence signals considered in the analysis of this study, for example when binning the data based on the running period? This issue will be particularly relevant when the mice change their running speed during the baseline or visual stimulations.

We completely agree with this comment. In addition to generally improving the accuracy of our wording throughout the paper, and removing any references to firing rate, we wish to also further clarify the way we deal with calculating neural tuning curves to running during visual stimulus presentation (figure 1). Hopefully this clarification addresses the specific concerns regarding the decay time of GCaMP potentially influencing our results.

We collect the trace (with $\Delta F/F$ defined with a rolling baseline, regardless of stimulus periods, as discussed above) for all periods during which there was any stimulus presented, and the corresponding speed at which the animal was running. We completely discard the ‘spontaneous’ period in between subsequent stimuli

presentations (in the few cases where such a period exists). This data plays neither into our 'spontaneous' results nor our 'stimulus evoked' results, in any way. We use a fixed offset of 66ms between the recorded neural data, and the stimulus presentation times / running speed measurements, to account for delays for the rodent visual cortex to receive the visual stimulus. Now, ignoring any stimulus identity, we bin and average the neural data according to 20 quantiles of the running speed distribution.

During this procedure, artefacts from the autocorrelation (decay time) of GCaMP could bleed in, in principle, if the animal changes his / her running speed frequently on time scales faster than 100 ms (e.g. faster than the decay constant). Our rationale for ignoring this artefact is that the autocorrelation time scale of running speed is far longer than 100 ms in this data. We have included an example autocorrelation plot of an animals' running speed from this dataset, where there is significant autocorrelation in the data on the timescale of 10's of seconds. Due to this relatively long timescale of running autocorrelation, the GCaMP autocorrelation can be safely ignored (e.g. animals change their running speed more slowly than GCaMP responds). Artefacts could also bleed into our selection of stimulus presentation periods vs. spontaneous periods, however in general those stimulus or spontaneous activity blocks correspond to at least many seconds to 10's of minutes of sequentially collected data, so such edge effects would only very minutely effect results, and in our opinion can be safely ignored.

c) The decoding analysis is based on the averaged df/f over 2s. It is unclear how the averaged df/f is calculated when the mice changed their running speeds during the stimulus presentation (or during the 1s baseline period). Are these trials discarded, and if so what are the proportions of the trials that are discarded in each mouse?

We thank the reviewer for pointing out this hole in our analyses! In our previous analysis, we did not consider potential dramatic changes in running speed during each 2s stimulus presentation, and simply considered the mean running speed for each trial. However, in order to control for any potential noise this assumption may have added to our analyses, we repeated our experiments with an additional control, as suggested by the reviewer.

In this control we excluded trials from the “running” segment if the minimum running speed dipped below 1 cm/ second (as well as if the average running speed was lower than 5 cm/ s), and excluded trials from the “stationary” segment if the maximum running speed was greater than 5 cm/ second (as well as if the absolute value of the average running speed was higher than .5 cm/ s).

Along with these requirements on running speed distribution, for each mouse, we also required there to be at least 50 stimulus presentations during the running period, and 50 stimulus presentations during the stationary period. These numbers were chosen to be the minimum constraint that allowed us to balance classes and stimulus presentation numbers between running and stationary periods. As a result, we had to exclude a fairly large number of mice from the decoding analysis – this was mostly because they either ran or were stationary for too large a portion of the drifting gratings presentation. The numbers are below.

VISpm (9 / 21 experiments used)

VISp (8 / 24 experiments used)

VISal(8 / 20 experiments used)

VISI (12 / 23 experiments used)

VISrl (3 / 6 experiments used)

VISam (7 /15 experiment used)

d) How was df/f defined when no stimuli were presented (e.g., “spontaneous” in Figure 1f and Supplementary Figure 1g)? If the baseline F_0 is the mean fluorescence of 1 second period preceding the no stimulus presentation, df/f following this baseline period would be around 0, regardless of the running speed.

We apologize again for the confusion regarding these spontaneous periods in between stimuli presentation. We did not use them. For our “spontaneous” conditions, we only considered experiments where there were long blocks without any stimulus presented - e.g. the 10 minute blocks of spontaneous activity recorded during each experiment.

In addition, as mentioned in response to a) we did not utilize the baseline period that the Allen institute used. We simply used raw $\Delta F/F$, binned according to running speed.

These pieces of information need to be described in much more detail in Methods. In addition, example traces of df/f would be helpful for the readers to capture what the data quality is and how the analysis was processed.

We sincerely apologize for the confusion due to the sparsity of our method sections. We have greatly expanded the methods section to address this concern.

We have now included example traces and cell filters in supplemental figure 6, which we hope will help the reviewers evaluate the quality of the data we are analyzing.

2) The relationship between visual responses and locomotion tuning

Despite that this study is on the neural coding in the visual cortex, there are no attempts to investigate the relationship between visual-response tuning vs. locomotion tuning. Do visual responses and the effect of locomotion add up linearly? Supplementary Figure 1e-g suggests that the effect of locomotion on spontaneous activity could be different from that on visual responses at the population level. This could imply complex effects of locomotion on visual responses. More detailed descriptions on the non-linear effects are necessary.

We appreciate the suggestion from all of the reviewers to further pursue the relationship between visual coding and running speed modulation. We originally were unsuccessful in pilot attempts to pursue this type of analysis, mostly due to the design of the Allen Institute dataset, which was not optimized for these types of analyses. In particular, most stimuli (like drifting gratings) were presented with barely enough repetitions to compute good tuning curves or response profiles (on a population or single cell level) even without splitting the data up into two different bins (running and stationary). Because of this, we were unable to address questions like “does the preferred stimulus of a neuron change during locomotion”, and changes in receptive field properties, etc. It will be very interesting for future work to pursue these directions, if more specialized datasets are collected from higher order visual regions with these analyses in mind.

However we agree with the reviewer that the supplementary data suggest some non-linearity in the interaction between running modulation and visual stimulus presentation. While we did not specifically further evaluate the correlates of locomotion in the spontaneous data, we did perform an additional analysis which we think begins to speak to the question the reviewer is raising. In an effort to further understand the relationship between stimuli encoding and locomotion, we added a new data figure to the main paper, and associated discussion in the results section. It is reproduced subsequently. We hope these analyses and new discussion address some of the reviewers' curiosities!

“We were curious to further investigate the interaction between neural correlation to running, and responses to visual stimuli. To do this, we restricted our analysis to neurons who responded significantly to drifting grating stimuli (this stimuli was chosen because it strongly drives neurons across all of the visual areas investigated here). We first repeated our previous analyses on these neurons, with qualitatively similar results (Figure 2a, b). Indeed, we found no correlation between the selectivity of each neuron, and the correlation coefficient between that neuron and the running speed (fig 2f-k). We

next analyzed the correlation between neural activity and locomotion only during presentation of either that neurons preferred or anti-preferred stimuli, in neurons that were strongly tuned to drifting gratings (fig 2c-e, drifting gratings selectivity index $> .95$). Interestingly, for neurons with an overall negative correlation to running, that negative correlation was less strong during the preferred stimulus than during the anti-preferred stimulus, indicating its ability to respond to its preferred stimuli is relatively retained. Unfortunately, the Allen data was insufficient to determine whether preferred stimuli changes during locomotion. Further data must be collected in order to determine if this holds in the higher order visual regions examined here.”

Figure 2 a) Fraction of neurons displaying different tuning types to running, when only considering neurons significantly responsive to the drifting gratings stimulus. b) Average correlation coefficient to running, computed during the presentation of the drifting gratings stimulus. Only includes neurons significantly responsive to drifting gratings stimuli. c) Average correlation coefficient to running calculated during all gratings, preferred grating orientation, and anti-preferred orientation. Only including neurons with an overall positive monotonic correlation to running. d) Same as c. except only considering neurons with an overall negative monotonic correlation with running. e) Same as c. except only considering neurons with non-monotonic tuning to running. f-k) Scatter plots comparing the drifting gratings selectivity index and correlation coefficient to running in each visual region.

3) Tuning to the running speed

The authors showed example cells that are tuned to specific running speed at the top of Figure 1b, and yet, used Spearman’s rank correlation coefficient to capture the population trend. The

correlation coefficient of the cells like those at the top of Figure 1b can show strange values. It is necessary to distinguish the cells that are monotonically increasing/decreasing and the cells that are tuned to specific running speeds more carefully.

We partially agree that using spearman's correlation coefficient could, in principle, lead to some cells that are 'tuned' to running in a more complex way than 'monotonic positive' or 'monotonic negative' being possibly miscategorized in our current analyses. However, it was always our intention that those neurons in the first row of Figure 1b belong to a 3rd class of tuning, "non-monotonic but never-the-less tuned". We categorized these cells by first finding all the cells with significant tuning to running (by comparing whether the variance of the binned tuning curve was statistically higher than the variance of a tuning curve calculated with shuffled running speed indices), and then 'removing' those cells whose firing was statistically significantly monotonic. In the population data of figure 1, we are only averaging over cells whose firing is statistically significantly monotonic (indeed, for the rest of the paper as well we only consider these cells). We absolutely agree that neurons who display tuning that is non-monotonic are extremely interesting, however they represent a relatively smaller fraction of the population, which is consistent across all the visual regions we analyzed - we now include in the main figure 1 a panel which shows these fractional distributions. In this work we focused on just the bulk trends towards positive and negative correlations, as opposed to honing in on specific details of the locomotion tuning (which would be very interesting for future work!).

Nonetheless, we wished to make sure that those neurons whom we classified as monotonic were not displaying "strange values" (although we note neither of those cells has a significant spearman's correlation coefficient). We did further analyses following Saleem et. al. (2013), where we fit 3 models to each cell that was significantly tuned to running (by the previous statistic).

As in Saleem et. al. the three different models were Gaussian functions, differentiated by constraints on the locations of the center parameter. The first model was a monotonically increasing model where we constrained the center of the Gaussian to fall higher than the highest running speed, the second model was a monotonically decreasing model where we constrained the center of the Gaussian to fall lower than the lowest running speed, and the third model was one where the center was allowed to fall within the highest and lowest values. We determined which model fit the data best 10 repetitions of a cross validation procedure where we split the data into 75% training, 25% test set before creating the tuning curves that we fit the models too. We have included these new analyses as part of supplemental figure 1, included below. This model fitting procedure qualitatively agrees with our non-parametric approach. We

agree that it's important to highlight the presence of cells with bandpass, stopband, or other interesting tuning in the paper.

Supplemental Figure 1 **a.** Average correlation coefficient for neurons monotonically tuned to running determined to either be band-pass, positively tuned to running, or negatively tuned to running based on fitting of a constrained Gaussian function (see methods) **b.** Fraction of neurons displaying different tuning types to running, with tuning type calculated via model fitting as in a. **c)** Ratio of neurons displaying increasing and decreasing tuning to running, calculated via model fitting as above. **d.** Correlation coefficient between pupil diameter and calcium activity in monotonically correlated neurons. **e.** Average correlation coefficient with running for monotonically tuned neurons, individual cre-lines displayed, calculated across all stimuli types. **f.** Pearson's correlation coefficient between running speed and dF/F in each region, calculated only natural stimuli (natural scenes, natural movies) were displayed. **g.** Same as e. but calculated only when synthetic stimuli (e.g. gratings, noise) were displayed. **h.** Number of tuned neurons displaying monotonic increasing vs. monotonic decreasing tuning to running, split out by natural and artificial stimulus types. **i.** Number of neurons tuned to running, split out by periods with and without stimulus.

4) Leaky integrate and fire model

a) The Methods do not contain essential information related to the model. For instance, the equation is missing, and it is not clear how the “input current drawn from Gaussian shaped tuning to 15 different objects” are related to the current range described in the manuscript

We apologize for the confusion! The input current is exactly the value given by the Gaussian tuning curve - e.g. the x axis of the Gaussian function is object identity, and the y axis is current value. When we varied the “gain” of each neuron to change its reliability, we changed the peak height of the Gaussian function. We swept between a value that produced essentially no firing (peak input current of .5nA) to value that produced a high firing rate (5 nA). We have added a schematic to the main figure which hopefully aids in the interpretation of those results, as well as included more details in the methods section of the main paper. The schematic is reproduced below, and shows an example of the current (corresponding to presentations of different ‘stimuli’) that was injected to an LIF neuron, and the response of that neuron.

The new Methods section is as follows:

To examine the possible effects of membrane voltage fluctuation on response reliability, we performed simulations of leaky integrate and fire neurons, using Euler's method. Membrane voltage was initialized to 70mV, and then integrated over time according to the following differential equation:

$$C_m \left(\frac{dV}{dt} \right) + g_l(V - E_l) = I_{app}$$

$$\text{If } V_{mem} > V_{thresh}, V_{mem} = E_l$$

C_m: membrane capacitance

g_l: leak conductance

E_l: resting membrane potential

I_{pp}: applied current

V_{thresh}: Threshold voltage

The following parameters were used: V_{thresh}: -49mV, V_{init}: -70 mV, integration time step = 0.05 ms, C_m = 4.9 ms, g_l = .16 us, E_l = -65. For Supplemental Figure 4, we did an additional simulation with resting membrane potential (E_l) of 68mV and 70mV for running and stationary periods respectively, to match the resting membrane potentials measured in vivo^{4,9}.

We created a Gaussian shaped tuning curve across 15 different stimuli to define the input current generated by each stimulus (as illustrated in figure 4). We simulated different levels of membrane voltage fluctuation by adding independent Gaussian noise to membrane voltage at each time step. Noise variance levels of 19(mV)² and 36(mV)² were empirically determined to match values recorded in vivo for running and stationary animals. We simulated presenting 300ms trials of each stimulus 10 times in a randomized order, and calculated reliability in the same fashion as described above.

b) The model assumes that the visually evoked currents are exactly the same between stationary and running periods, but this assumption would not hold for the neurons in higher-order visual cortex, as they receive inputs from neurons in V1 (that has higher reliability during locomotion).

That's definitely true, and would only aid in achieving our desired effect (high reliability despite lower firing rates). Nevertheless, our simulations are meant to show that, *all else*

being equal, simply the changing background noise can cause all of the effects we observe in our data. (E.g. would, in the absence of other factors, both decrease firing rates and increase response reliability, where for moderately strong input currents (or similarly strongly tuned neurons), the decrease in firing rate would be spared for that neuron's most preferred stimuli. This is because the input current would already easily drive the neuron above its threshold voltage, and the reduction in random fluctuations would not significantly decrease the number of spikes emitted by the neuron.

c) Does this mode assume that the baseline membrane potential is the same during running and stationary periods? Is this assumption consistent with the data in Polack and Golshani, 2013 (from which the parameters of the model derive)?

We appreciate the reviewer bringing this point to our attention. In principle, the difference in membrane voltage and the difference in membrane noise could be due to differing mechanisms that also contribute differently to neurons that are negatively correlated with running vs. positively correlated with running. This is actually our personal hunch for the mechanism behind our findings, and the reason we wished to consider the difference in membrane fluctuation noise in isolation from other factors. Nevertheless, we repeated the simulations including the membrane potential differences as observed in both the Hestrin¹ and Golshani² papers, and we still observed that change from the stationary to running membrane properties lead to an increased reliability, in certain input regimes. The figure is reproduced below, and now is supplemental figure 4d. As an aside, it would be interesting if someone were to repeat these experiments and differentiate between arousal and locomotion per se (ala Cardin et. al³), or different cortical regions or neural tuning types.

5) Relationship between pupil size and neural activity

One of the popular ways to assay the “arousal” (page 2) level is to measure the pupil diameter, which has also been employed in the study of the primary visual cortex (e.g., Vinck et al., 2015; Reimer et al., 2014). Since “Eye movements and locomotion speed were recorded” (Methods),

I would expect analysis on the relationship between the pupil diameter and visual responses included in this manuscript.

Absolutely, we are very glad to be able to include this analysis in the revised manuscript. At the time of submission, the Allen Institute had not yet publicly released the pupil area information, although it is now available. We re-ran all of the analyses for figure 1 using running speed as a regressor instead of pupil diameter. The main result has been added to supplementary figure 1, and is unsurprisingly basically identical to that arrived at when using running speed in the main manuscript. We realize that locomotion and pupil diameter are highly correlated variables – and although we think it is an extremely interesting and worthy area of study to try to decouple their effects on neural coding, unfortunately that is outside of the scope of this manuscript. This is mostly because the data were not collected with this type of analysis in mind, there are very few periods where the mice are naturally aroused but not running in the data, unfortunately not enough to draw any sturdy conclusions from. Instead our data are best interpreted from the perspective of locomotion as an assay of arousal state, or broad behavioral state, as opposed to being about locomotion per-se. We have clarified this in the introduction, and added new discussion points regarding the distinction between locomotion and arousal.

6) Some of the descriptions in the introduction and the discussion need improvement.

a) In page 1, “However, the investigation of these effects has so far been limited to primary visual cortex (V1).” This statement is wrong. The effect of locomotion on visual responses was investigated in areas AL and PM in a pioneering work by Andermann et al., 2011. In fact, Andermann et al reported no significant suppression in PM, which contrasts with the current study. This issue needs to be discussed in detail. To find out the discrepancy, it is advisable to analyze the Allen database in the same way as Andermann et al.

We appreciate the reviewer for reminding us of the Anderman et. al. study! We neglected the treatment of running speed in that paper, because it was mainly used as a control for differences in selectivity distribution they found between different visual cortical areas. They found the **peak** neural response was slightly increased in AL, insignificantly increased in PM (using data from 27 and 8 neurons, respectively), which is completely compatible with our findings, especially considering our finding in our new Figure 2. We do find that in AL, amongst neurons that are monotonically correlated with running, the net modulation is slightly positive, although less so than V1. In PM, we find the net modulation to be essentially zero (e.g. split between positively and negatively correlated cells). In addition, amongst cells that **are** negatively correlated with running, their peak response – the response to their preferred stimulus, was less negatively correlated with running than when the correlation was calculated when all

data were considered. We have also changed the wording of the abstract and introduction to be more sensitive to previous and new work addressing the impact of locomotion on regions other than V1.

“Running profoundly alters stimulus-response properties in mouse primary visual cortex (V1), but its effect in higher-order visual cortex is relatively unexplored. Here we systematically investigated how visual responses change during locomotive state across six visual areas and three cortical layers using a massive dataset from the Allen Brain Institute. Although running has been shown to be primarily positively correlated with neural activity in V1, we found many neurons whose activity was negatively correlated with running in extra-striate regions. Nevertheless, across all visual cortices, stimuli presented during running could be decoded from the neural responses more accurately than visual stimuli presented during stationary periods. We show that this effect was not attributable to changes in population activity structure, and propose that it instead arises from increased reliability of single neuron responses during locomotion.”

To understand perception, it is important to study how contextual variables (like behavioral state) affect the representation of sensory information in neural populations. Locomotion, a highly ethological behavior in rodents, is accompanied by pronounced changes in the magnitude and consistency of neural responses to visual stimuli¹⁻⁹. However, the investigation of these effects has so far been largely restricted to the primary visual cortex (V1).”

b) It is not clear how the second hypothesis described in the Introduction can be addressed by Allen’s datasets. Could this be clarified?

In retrospect, we agree with you that the hypothesis about predictive coding cannot really be addressed using the Allen institute dataset. We have removed that sentence from the introduction.

Minor:

1) Please include the number of cells in the figure legends.

Thanks for this suggestion. Each separate animal / recording session was considered separately for our decoding analysis, as we wished to assess the contribution of the population correlation structure to the decoding performance. Given we analyzed multiple mice from each of 7 visual regions, we think listing all the cell numbers in the figure legends might get a bit unruly. We have instead created a supplemental table

which lists cell numbers for each separate mouse / experimental session, and visual region. We hope this is acceptable!

2) “average fluorescence (df/f)” (for example Figure 1b)

df/f is not fluorescence, but an index normalized to the fluorescence at the resting state to capture the neural activity.

Thanks for catching this, fixed in the figure, and we now refer to either “normalized fluorescence (df/f)” or just df/f in the paper.

2) P3: Figure 1c. “dff” should be “df/f”.

We fixed these issues, thanks!

4) P4: “These trends were consistent across multiple choices of classifiers (Supplemental figure 2)”

Supplemental Figure 2 shows the results for one classifier. Did the authors test only two classifiers in total?

We also tested multi-class SVM, the results of which are now presented as a new supplemental figure.

5) P4: “reduced noise curious could”

“curious” should be “correlation”

6) P5: (figure 2k). This should be figure 3d.

7) P6: mV^2 should be represented as $(mV)^2$

8) P6: (figure 3k). This should be figure 3e-g.

We appreciate the minor comments and have fixed them in the manuscript!

Reviewer #2:

This manuscript, by Chistensen and Pillow, represents a very new kind of neuroscience. In this case, the authors have sourced a primary data set from the Allen Institute Brain Observatory, a repository of 2-photon imaging data collected in the visual areas of awake mice. Using these data, they have investigated the relationship between locomotion and firing rates in primary and higher-order visual cortices. The question of state-dependent regulation of neural activity and gain control is a central one in the field, and could potentially be very interesting as a starting point for examining the flow of information among higher-order cortical areas. However, the authors have not addressed this issue in a manner leading to a clear set of

answers. Indeed, it is uncertain that the dataset they have chosen would allow any such clarity, due to inherent restrictions. Furthermore, the nature of the dataset precludes addressing these problems by changing the experimental paradigm, and I do not see any way for the manuscript to be substantially improved under these conditions.

Overall, the authors seem to be mistaking ‘modulation of firing rate by running speed’ for ‘locomotion-induced gain modulation of visual responses’. Firing rate alone is not interesting, and the experimental paradigm employed here is not optimized to answer the main questions posed by the authors. My main concerns are outlined in more detail below.

First, because there are still few or no papers published from the Allen data set, the onus is on the authors to perform appropriate due diligence. It is not sufficient to reference the Allen white paper. The authors need to provide some quantitative assessment of the quality of the data. In fact, the authors explicitly state that they did no post-processing of the data once downloaded, suggesting that ROIs were not examined for overall activity, stability of activity over time, or visual responsiveness. Furthermore, the authors present neither traces from individual ROIs nor statistics on ROI quality or criteria for inclusions—all elements that would allow the reader to evaluate the quality of the data set.

We thank the reviewer for pointing out these holes in our original submission! Fundamentally, we were trying to avoid presenting too much about the data collection methodology that we did not ourselves perform in the data figures of our paper. However, it is clear from the responses from multiple reviewers that we erred by going too far in this direction, making our analyses difficult to interpret. We absolutely did visually inspect the traces and ROIs we included in the analysis to ensure their quality. To address these concerns, and allow the readers to do the same, we have included a new supplementary figure (supplemental figure 5) showing traces and ROIs from example brain regions and cre-lines in the dataset, also included below. It is our opinion that these traces and ROIs are qualitatively very good.

Importantly, the peer reviewed paper describing the Allen Institute Brain Observatory dataset is now published in Nature Neuroscience (de Vries et. al. 2019), this work describes the data collection methodology in detail and includes substantial quantitative validation of the data. To address the reviewers specific concerns regarding stability over time, visual responsiveness, and general methodology:

1. The results of all figures are qualitatively unchanged if we **exclude** all cells from the database that are not visually responsive (as determined by significantly responding to the drifting gratings stimuli which we used for decoding) - this data is now shown in our new figure 2, included below.

2. The results of all figures are qualitatively unchanged if we consider cells that are matched across days / experiments as one cell, or if we consider each day / experiment separately and perform no cell matching (data in the current version of the paper retain cells as matched across experiments, while the previous version considered each experiment as independent). Cell traces that were matched across days were visually inspected for gross changes in SNR, etc. We found the rolling baseline method of calculating dff employed by the Allen Institute controlled for this issue very well.
3. We have included substantially more methods about the data treatment, etc.

We hope the above clarifications and additions to the manuscript (especially the peer review and publication of the primary paper describing the Allen Institute dataset) help the reviewer to better evaluate the quality of the data - and also to feel comfortable that we did due diligence in our evaluation!

New Figure 2 and Related Discussion:

“We were curious to further investigate the interaction between neural correlation to running, and responses to visual stimuli. To do this, we restricted our analysis to neurons who responded significantly to drifting grating stimuli (this stimuli was chosen because it strongly drives neurons across all of the visual areas investigated here). We first repeated our previous analyses on these neurons, with qualitatively similar results (Figure 2a, b). Indeed, we found no correlation between the selectivity of each neuron, and the correlation coefficient between that neuron and the running speed (fig 2f-k). We next analyzed the correlation between neural activity and locomotion only during presentation of either that neurons preferred or anti-preferred stimuli, in neurons that were strongly tuned to drifting gratings (fig 2c-e, drifting gratings selectivity index > .95). Interestingly, for neurons with an overall negative correlation to running, that negative correlation was less strong during the preferred stimulus than during the anti-preferred stimulus, indicating its ability to respond to its preferred stimuli is relatively retained. Unfortunately, the Allen data was insufficient to determine whether preferred stimuli changes during locomotion. Further data must be collected in order to determine if this holds in the higher order visual regions examined here.”

Figure 2 a) Fraction of neurons displaying different tuning types to running, when only considering neurons significantly responsive to the drifting gratings stimulus. b) Average correlation coefficient to running, computed during the presentation of the drifting gratings stimulus. Only includes neurons significantly responsive to drifting gratings stimuli. c) Average correlation coefficient to running calculated during all gratings, preferred grating orientation, and anti-preferred orientation. Only including neurons with an overall positive monotonic correlation to running. d) Same as c. except only considering neurons with an overall negative monotonic correlation with running. e) Same as c. except only considering neurons with non-monotonic tuning to running. f-k) Scatter plots comparing the drifting gratings selectivity index and correlation coefficient to running in each visual region.

Example ROIs:

VISp Cux2-CreERT2

VISI Cux2-CreERT2

VISal Cux2-CreERT2

Supplemental Figure 5. Example cell traces and ROIs. Colors in cell mask ROI images (left) correspond with colors in cell trace df/f images (right).

A number of key points about the analyses and the data are missing from the manuscript, leaving the interpretation of the data very unclear. What method for dF/F was used in the final analysis? How was the baseline calculated? Several methods for baseline estimates have been used consistently in the field, including normalizing to the mean across an imaging session and normalizing to the lowest 10% of the values. However, these methods affect the range of values expected for spontaneous and evoked activity. Along similar lines, what percentage of cells were deemed visually responsive, according to what criteria?

We apologize for these oversights in the methods! We have expanded the methods section greatly, following comments from all the reviewers. As mentioned in the response to reviewer 1 the df/f method used was a rolling baseline, as described in the response to reviewer one, and new details in the methods section (reproduced here).

“ $\Delta F/F$ was calculated using a rolling-window baseline method. The baseline was calculated as a windowed mean of the windowed mode. The mode kernel width is 5400 frames (3 minutes), and the mean kernel width is 3000 frames (1.667 minutes). The raw, neuropil-corrected fluorescence was then baseline-subtracted and baseline-normalized.”

We did not originally sub-select neurons that were visually responsive, as we marginalize over all stimuli presented to the animal to calculate the correlation between neural activity and running speed. However, in the new figure 2 we reproduced some of the panels from figure 1 including only neurons that were responsive to drifting gratings (with qualitatively similar patterns as the overall population). In addition, figure 2 further explores the relationship between visual tuning and correlation to locomotion.

Presumably the mice are running in lighted conditions, since visual stimuli are shown, but this is not stated. Using data from light-only periods restricts the authors' ability to compare their findings with other groups', since recent work has distinguished locomotor responses in local V1 circuits occurring during dark and light running periods. Comparing locomotion-associated gain changes in dark and light conditions would enhance the authors' ability to draw comparisons between V1 and higher-order areas.

The spontaneous data were collected during periods of mean luminance grey visual stimulus. These data were blocks of 10 minutes included in every Allen institute experimental session. Although we present the data from spontaneous periods in the main manuscript for those that are curious, comparison between stimulus and spontaneous periods is not the main story we wished to pursue.

It is unclear which periods the ‘spontaneous’ data are taken from. Because the Allen white paper suggests continuous visual stimuli, 2s on and 2 s off, the ‘spontaneous’ data are presumably taken from the inter-stimulus intervals. Even for gCAMP6f, which has relatively rapid kinetics, this means that several hundred milliseconds of the ‘spontaneous’ intervals between visual stimuli are in fact contaminated with the offset tail of the calcium signal. This makes the comparison of the ‘visual’ and ‘spontaneous’ datasets problematic. None of the necessary controls or data are presented to address this issue. The authors should instead present stimuli with > 1 s between stimuli to reduce the impact of this issue.

The spontaneous data are taken from the 10 minutes during each experiment where no visual stimulus was displayed (during this time a mean luminance white screen was displayed). We discarded the inter-trial interval entirely. We apologize for not explaining this point clearly. We have clarified this in our methods section.

The authors suggest that the main point of the study is to examine the ‘prevalent view that running acts to enhance visual representations,’ but none of the analyses directly address visual representations at all. The issue of locomotion-induced changes in visual gain is quite complex, because the locomotion-associated increase in firing rate observed in V1 and the change in visual SNR are potentially mechanistically distinct. Here the authors have calculated only dF/F measurements, as a proxy for firing rate. Firing rate is not sufficient to address modulation of visual gain and encoding, and the authors should instead calculate a measure of SNR or visual response amplitude. In addition, the analysis of visual responsiveness should be restricted to cells that show a significant modulation by visual stimuli under at least one condition—up to 25% of V1 cells may not respond to visual stimuli under these experimental conditions.

We thank the reviewer for these comments as they motivated us to include the new figure 2. We would like to separately address two points: 1. the concept of “gain” in general, and 2. the interaction between locomotion and “visual representations”.

1. It is clear to us from reading the reviewer comments that the word gain often has a specific meaning, sometimes with mechanistic implications, which we did not intend on invoking. We no longer refer to running speed modulation of Ca^{2+} activity as “gain” in the manuscript.
2. We have now included a new figure 2, in which we evaluated whether locomotion tuning changed as a function of visual stimuli, as suggested by the reviewer. In particular we evaluated running speed correlation for a subset of neurons that were significantly direction selective, only during presentation of

drifting grating stimuli. For this significantly reduced dataset, the overall pattern of Ca²⁺-running speed correlation was consistent with what we observed in the entire dataset. In particular the average correlation was positive in V1, LM and AL, whereas it was negative in AM, RL and PM. We further investigated how neural activity was correlated with running speed during presentation of the preferred and anti-preferred stimuli of each direction selective neuron. We were particularly interested in trying to determine how the tuning curve of neurons whose overall activity decreased during locomotion changed. We found that the activity predominantly reduced during presentation of the non-preferred stimuli, leaving the activity intact during the preferred stimuli. This effectively increases the SNR of these neurons, despite their reduced overall activity! These results are actually quite consistent with the regime predicted by our simulations that would support increased reliability despite decreased overall firing rates.

3. As a minor point, we disagree with the reviewer that df/f measurements, marginalized across a many visual stimuli aren't sufficient to address locomotion induced modulation of visual encoding. In our analysis, we compare distributions of df/f values across two sets of visual stimuli, from stationary and locomotive periods. These two sets of visual stimuli are both drawn from the same distribution, and we show that there is a statistically significant change in neural activity between these two different sets of stimuli; significance here is established by comparing to a null distribution created by shuffling the animal's running speed with respect to the neural response data. Importantly, this could only happen if neural responses to stimuli differ conditioned on locomotion speed. We apologize if we have misunderstood the reviewers' critique, and would be happy to explore the issue further if the reviewer feels there is a problem with our argument.
4. As mentioned previously, we have now done analyses including only visually responsive neurons.

A related issue is that the spontaneous and visual datasets are conflated, and the visual stimulus properties are constantly varying. This presents a challenge for calculating SNR. A better experimental paradigm would be to present fewer, or one, stimuli many times in succession to examine visual response modulation by running. Alternately, the authors might concentrate on orientation tuning or contrast gain.

We thank the reviewer for this comment, as it is clear we did not adequately describe our methods on this point. As we clarified in more detail earlier in our response to this reviewer, we did not use the 'spontaneous' periods collected within the visual datasets. Indeed, as described in the previous response, we have indeed focused on responses to drifting gratings, conditioned on running speed, to further investigate the statistical dependence of visual coding on locomotion.

Finally, as a general response to the points raised by this reviewer, we wish to clarify 1) we are considering responses to **visual stimuli** during running and not during running. We observed that visual responses during running were LOWER than visual responses during stationarity, in some visual regions. Part of the reason we emphasized this finding is that it is unexpected given previous findings in the literature, which have generally demonstrated higher firing rates during running. 2) However, we have made the point even clearer by re-analyzing responses to individual stimuli both during running and during stationarity, and now show that responses to the SAME STIMULI decrease during running (in some higher-order regions). Our main point is that despite the fact that increase in neural activity during locomotion is not a generalized visual cortex phenomenon, increase in coding accuracy **is**. We hope that this addresses the reviewer's concern, though we apologize if we have misunderstood the reviewer's original comment.

Reviewer #3:

In this manuscript, the authors characterize how locomotion affects neural activity in different areas of mouse visual cortex. They find running-related fluorescence increases in V1 and two other areas, and decreases in AM. Two other areas (PM, RL) have smaller decreases. They use decoding approaches to study which factors (amplitude, correlation, reliability) control changes in information about visual stimuli between running and stationary conditions. The paper is well-written and clear.

The question of how running changes information content is of general interest. However, there are several concerns that cast doubt on the findings as stated.

Major:

1. The abstract says "... suppressed firing in higher-order visual areas. Despite this reduction in gain, visual responses during running could be decoded more accurately than ... during stationary periods." Figure 1c seems to show that LM (VISI), and AL increase fluorescence, contrary to the abstract claim, and only AM shows a substantial decrease during running. And Fig. 2b does not provide clear evidence that AM (pink/magenta) shows an improvement in decoding performance during running. The work seems to support instead the

idea that V1 increases firing rate with running, that V1 carries the largest amount of information about the stimulus, and that V1 information is increased the most by running (Fig. 2). Perhaps additional analyses, focusing on AM and examining visual gain in more detail, could provide support for the present claim, but it seems that a major rewrite would be required.

We thank the reviewer pointing out the difficulty in interpreting our figures. We have redone the way we present the information in figure 2, and now it should be clear that although it is indeed the case that V1 has the largest decoding accuracy in general, in all areas the decoding performance increases during locomotive periods.

We have also changed the wording of the abstract and main paper to be more cautious, we intended to convey only that the running induced gain is the largest in V1, and decreases the further anterior you go in the striate cortex, as shown in figure 1g. We changed the wording of the paper and abstract to reflect our intention of considering correlation trends **relative** to V1.

We included substantial additional analyses relating to how the coding of visual stimuli interact with locomotion in Figure 2. We hope that this will be of interest to the reviewer, we have included it below.

“We were curious to further investigate the interaction between neural correlation to running, and responses to visual stimuli. To do this, we restricted our analysis to neurons who responded significantly to drifting grating stimuli (this stimuli was chosen because it strongly drives neurons across all of the visual areas investigated here). We first repeated our previous analyses on these neurons, with qualitatively similar results (Figure 2a, b). Indeed, we found no correlation between the selectivity of each neuron, and the correlation coefficient between that neuron and the running speed (fig 2f-k). We next analyzed the correlation between neural activity and locomotion only during presentation of either that neurons preferred or anti-preferred stimuli, in neurons that were strongly tuned to drifting gratings (fig 2c-e, drifting gratings selectivity index > .95). Interestingly, for neurons with an overall negative correlation to running, that negative correlation was less strong during the preferred stimulus than during the anti-preferred stimulus, indicating its ability to respond to its preferred stimuli is relatively retained. Unfortunately, the Allen data was insufficient to determine whether preferred stimuli changes during locomotion. Further data must be collected in order to determine if this holds in the higher order visual regions examined here.”

Figure 2 a) Fraction of neurons displaying different tuning types to running, when only considering neurons significantly responsive to the drifting gratings stimulus. b) Average correlation coefficient to running, computed during the presentation of the drifting gratings stimulus. Only includes neurons significantly responsive to drifting gratings stimuli. c) Average correlation coefficient to running calculated during all gratings, preferred grating orientation, and anti-preferred orientation. Only including neurons with an overall positive monotonic correlation to running. d) Same as c. except only considering neurons with an overall negative monotonic correlation with running. e) Same as c. except only considering neurons with non-monotonic tuning to running. f-k) Scatter plots comparing the drifting gratings selectivity index and correlation coefficient to running in each visual region.

2. The authors use the word “firing” in the abstract, but relating calcium responses to firing can be difficult. The conclusion about reliability -- that increased df/f reliability implies increased spiking reliability -- is especially fraught. Changes in intracellular calcium concentrations and/or buffering could underlie changes (as just one example, perhaps ACh input changes calcium concentrations during running). GCaMP could also cause increased reliability, for example if the fluorescence response is moved closer to saturation by an experimental condition. It might be possible for the authors to support the claim that fluorescence reliability change is due to spike reliability change, but it seems difficult.

We agree with the reviewer’s point, and apologize for the imprecise use of terminology. We have changed our wording to indicate that we are always referring to either calcium fluorescence or ‘neural activity’, as opposed to neural firing. We also take the reviewer’s point that there might be important differences between fluorescence reliability and

spike reliability, and we have added this important caveat to the Discussion, included below.

An important caveat to our work is that although we have only measured and analyzed Ca²⁺ fluorescence as a proxy for neural activity, many previous theoretical studies on neural tuning, neural response gain, and even our own work on neural reliability through LIF simulations relied on analyzing neural firing rates. Ca²⁺ activity in many cases may be a noisy, non-linear proxy for actual firing rates, and further work should be done to verify these findings with electrophysiology.

Minor:

- It was difficult to determine whether changes in response were calculated during spontaneous periods, or in response to visual stimuli (as is most appropriate when “gain” is used). The df/f changes from Supp. Fig. 1 should be in the main text. And do results change if exclusion of excited neurons is based on artificial/natural/spontaneous responses?

We apologize for the confusion! We have treated spontaneous periods and response to visual stimuli separately in the manuscript, and the main results are all based on locomotion-Ca²⁺ correlation calculated during periods of visual stimulation. As suggested by multiple reviewers, we have expanded the details of our methods section explaining how we deal with spontaneous / visual stimulation segregation.

As far as the word “gain” goes, we now generally try to avoid it, as it is mostly inappropriate for the analyses we have done. However the new figure 2 we have added, where we address how locomotion tuning interacts with tuning to drifting grating stimuli might be interesting to the reviewer.

The results are generally insensitive to whether exclusion of excited neurons is based on any of the criteria mentioned.

- Response gain should be shown by plotting response to the same visual stimuli in running and stationary states both at the single-neuron and population level.

We have tried to do an analysis in this fashion (at least at the single cell level) in the new figure 2.

- The supplementary materials should include the equations of the LIF model and the manuscript should justify leaving out recurrent responses to input, which seem likely to influence spike responses.

We now include the LIF equations, thank you! Similarly to the comment from Reviewer 1 regarding inputs from lower order regions impacting the firing in higher order regions, our simulations are meant to show that, *all else being equal*, simply the changing background noise can cause all of the effects we observe in our data (e.g. would, in the absence of other factors, both decrease firing rates and increase response reliability). We have added this caveat to our section explaining the LIF simulations, the relevant paragraph is included below:

There are many important features of actual cortical circuits (such as recurrence between visual areas, and even the fact that higher order visual areas receive input from lower order visual regions, which may already have running speed modulated activity), which are not included in these simulations. These simulations are intended to show that all else being equal, simply the changing background noise can cause all of the effects we observe in our data (e.g. would, in the absence of other factors, both decrease firing rates and increase response reliability). Further experimentation and physiological measurements will be required to establish whether a reduction in membrane voltage fluctuations during locomotion explains the enhancement in decoding performance we observed, however our simulations are consistent with this hypothesis.

Reviewer #4:

The authors explored the Allen Brain Institute database to discover that, during locomotion, the firing rate in V1 increases (consistent with previous reports) but, surprisingly, it suppresses responses in higher visual areas.

Despite this decrease in firing rate, the reliability of the responses increased, allowing a more accurate decoding of the visual stimulus.

Moreover, the authors show that a simple, leaky integrate and fire neurons may replicate the observed phenomenon if reductions in noise are allowed to counteract the decrease in firing rate during locomotion.

Major comments

One major concern is that most analyses are performed at the population level, in contrast with past studies that investigated how the tuning curves of individual neurons change between rest and locomotion. These data appear available from the Allen dataset. The study could be improved substantially by describing and modeling the changes in tuning across each of the areas to better understand how changes in gain and mean response varies as a function of behavioral state.

We thank the reviewer for this comment. To address it, we have included a new figure 2 where we focused on single cell responses to drifting grating stimuli. This analysis allowed us to identify differences in correlation to running speed in epochs when a neuron's preferred stimuli was being presented, vs. its anti-preferred stimuli. In particular, activity during the non-preferred stimuli seems to be the primary contributor to the negative correlation to running we previously observed. The new figure and related discussion is reproduced below.

“We were curious to further investigate the interaction between neural correlation to running, and responses to visual stimuli. To do this, we restricted our analysis to neurons who responded significantly to drifting grating stimuli (this stimuli was chosen because it strongly drives neurons across all of the visual areas investigated here). We first repeated our previous analyses on these neurons, with qualitatively similar results (Figure 2a, b). Indeed, we found no correlation between the selectivity of each neuron, and the correlation coefficient between that neuron and the running speed (fig 2f-k). We next analyzed the correlation between neural activity and locomotion only during presentation of either that neurons preferred or anti-preferred stimuli, in neurons that were strongly tuned to drifting gratings (fig 2c-e, drifting gratings selectivity index > .95). Interestingly, for neurons with an overall negative correlation to running, that negative correlation was less strong during the preferred stimulus than during the anti-preferred stimulus, indicating its ability to respond to its preferred stimuli is relatively retained. Unfortunately, the Allen data was insufficient to determine whether preferred stimuli changes during locomotion. Further data must be collected in order to determine if this holds in the higher order visual regions examined here.”

Figure 2 a) Fraction of neurons displaying different tuning types to running, when only considering neurons significantly responsive to the drifting gratings stimulus. b) Average correlation coefficient to running, computed during the presentation of the drifting gratings stimulus. Only includes neurons significantly responsive to drifting gratings stimuli. c) Average correlation coefficient to running calculated during all gratings, preferred grating orientation, and anti-preferred orientation. Only including neurons with an overall positive monotonic correlation to running. d) Same as c. except only considering neurons with an overall negative monotonic correlation with running. e) Same as c. except only considering neurons with non-monotonic tuning to running. f-k) Scatter plots comparing the drifting gratings selectivity index and correlation coefficient to running in each visual region.

There seems to be an interchangeable use of mean spike rates and gain in the manuscript that is somewhat confusing. For example, the authors write “Nevertheless, the observed suppressive effects of running on firing rates contradicts the naïve hypothesis that running induces selective attention to vision that increases the gain of responses throughout visual cortex”. However, the two are different concepts and mean spike rates and gain of responses may go in opposite directions (see discussion in Mineault et al, J. Neurosci, 2016). Changes in gain could be evaluated if the authors analyzed changes in the tuning curves of neurons during rest and locomotion, as suggested above. It is not clear how the estimated changes in gain from the present analysis. From the text, it appears they believe changes in gain and mean responses refer to the same phenomenon, but that is not how the terms are used in the literature.

Thanks for this comment. We have changed our wording to reduce this point of confusion, and now refer only to changes in the correlation between neural responses and running, as this is what we have measured, as opposed to changes in visual gain.

However, as we explained in our response to reviewer 2, in our analysis we compare distributions of df/f values across two sets of visual stimuli, from stationary and locomotive periods. These two sets of visual stimuli are both drawn from the same overall distribution, and we show that there is a change in overall neural activity across these two different sets of stimuli, by comparing to a null distribution created by shuffling the locomotion speed of the animal. Importantly this could only happen if neural responses to the individual stimuli were changing as a function of locomotion speed. So while we have removed the word “gain” from our paper as we did feel it was confusing in general, and our analyses do not follow the typical analysis used by visual neuroscientists when referring to gain, we do think we can *statistically* make the claim that there is an expected gain increase to the set of visual stimuli in our dataset.

The authors mention that decreases in noise in a simple LIF neuron could potentially replicate some of the phenomena observed, but they do not explain how this could happen. Given that V1 exhibits higher firing rates and increased reliability during locomotion, one would expect the input signals to higher visual areas to reflect this as well. Perhaps, the increased input rates engage local inhibition in a way that the overall mean rate in higher visual areas decreases, despite an increase in input. Once again, a cell-by-cell analysis of how tuning curves change between rest and locomotion may shed light into what is actually happening.

Overall, this reviewer felt the reported phenomena to be interesting, but that more analyses at the single-cell level are required to better understand how locomotion is changing the activity of neurons. Given that the data are available, it may not be very difficult for the authors to carry such analyses.

Minor comments

Fig 1. A population analysis of the tuning curves would be helpful. Are tuning curves about the same in all visual areas? Are some types (monotonically increasing/decreasing) predominantly found in some areas?

How was Fig 1d put together? Is this a single mouse?

We actually removed the figure the reviewer is referring to, as we felt it was too difficult to explain how it was made (it was actually multiple mice aligned to a common coordinate system). We feel the new figure 1g. is much easier to understand, and more illustrative. It is included below!

Figure 1 a. Schematic of regions included in study. b. Example non-monotonic, monotonically decreasing, and monotonically increasing tuning curves to running speed, top to bottom. c. Fraction of cells displaying different tuning types to running speed across the visual regions we examined. d. Average correlation coefficient between neural activity and running in each visual region, amongst neurons displaying significant ($p < 0.05$) monotonic tuning to running. e. Overall fraction of neurons significantly tuned to running in each region and Cre-line, calculated during visual stimulus presentation. f. Same as e. except calculated in the absence of visual stimuli. g. Visualization of spatial distribution of overall tuning to running in the visual regions.

Fig 2. How is decoding performance defined? I could not find this information in the “Decoding analysis” section. It would help to explain what is being plotted in the Fig legend for easy access.

Thanks for pointing out this omission. It’s simply the fraction of test stimuli correctly identified.

Fig 3. It would help to add a plot of reliability against noise level and mean response (instead of just the peak current). Mean firing rate will increase with both noise and peak current. This will help the reader evaluate better the claim that “that neurons had significant room to decrease their firing rates while still improving reliability.”

The main figure plot is now against mean input current (we agree this is more interpretable), while we have a supplemental figure with the peak input current figure (this corresponds to something like the neurons tuning gain).

Typo: “and reduced noise curious could” Remove “curious”

It would be useful to cite a relevant reference, Mineault et al, Enhanced Spatial Resolution... J Neurosci, 2016, which provides a framework to discuss how changes in gain and mean rate may relate to each other.

We thank the reviewer for pointing out the reference! It cemented our view that we were using the word gain in a way that is not quite commensurate with what is normal in the field. We realized in the first version of our manuscript we simply used the word “gain” to mean a statistical increase in neural activity across a distribution of stimuli (or in this case a change in the neural activity due to a set of stimuli, conditioned on running or not running) -- while common parlance in the field has a more specific definition about how neural activity changes to specific stimuli, compared to baseline. We have removed references to “gain modulation” from our verbiage in the new manuscript.

We do not yet know how attention modulates activity across different visual areas. Thus, it is not clear that the findings rule out increased changes in attention, but simply show that increased decoding accuracy does not necessarily need to be accompanied by increases in mean firing rate.

We agree with this comment and have added a caveat in the discussion to emphasize this point. The caveat is included below:

Similarly, our results could be a downstream consequence of a difference in distribution of preferred stimuli type across the difference visual regions – as previous authors have noticed that neurons with particular tuning preferences (namely those tuned to high spatial frequencies¹⁶) are more likely to be positively correlated with running. This effect has been described as a potential rodent analog of spatial attention. However this is unlikely to completely explain our results -- the published data on spatial frequency

selectivity in mouse higher order visual cortex¹⁴ does not correlate with the distribution of running speed selectivity we have observed. For example in Marshel et. al. AL has the lowest spatial selectivity, and PM has the highest spatial selectivity, whereas in our data AL and PM have relatively similar overall distributions of neural correlations with locomotion speed.

What were the distributions of spontaneous activity across all these areas? Is there a correlation between the spontaneous activity and the mean changes observed during locomotion across areas? Is there a correlation even within an area (perhaps cells with low spontaneous rate show increases while cells with high spontaneous show decreases)?

Due to the issues many reviewers have raised with the lack of a one-to-one relationship between firing rate and Ca²⁺ activity, it isn't completely straightforward to assess spontaneous firing rates comparatively between different brain regions and different animals, as baseline fluorescence and peak fluorescence levels (due to underlying spiking activity) are highly dependent on imaging conditions, expression levels, surgical quality, etc. In order to try to account for these issues but still address the reviewers questions, we felt the best course of action was to deconvolve (using the Oasis² package) the Ca²⁺ activity to binary spiking events, and then calculate an "activity rate" -- predicted spikes / second. This rate is unsurprisingly low, given Ca²⁺ deconvolution notoriously misses single spiking events. The average event rates in different regions, averaged separately for neurons with different tuning types, are plotted below. While there does seem to be a lower overall spontaneous activity rate in the higher-order visual areas, there is no consistent pattern between the different tuning types. E.g. a lower spontaneous activity does not predict that neurons will be classified as monotonically decreasing. It's possible lower "spontaneous activity" in higher order visual regions is due to their position on the outside of the imaging window, which would in general lower optical quality. However the fact that spontaneous activity is not predictive of tuning type (even amongst the most anterior higher order visual regions), comforts us that this same effect does not underlie the tuning type distribution we have observed in our analyses.

1. de Vries, S.E.J., Lecoq, J.A., Buice, M.A. *et al.* A large-scale standardized physiological survey reveals functional organization of the mouse visual cortex. *Nat Neurosci* 23, 138–151 (2020).

2. Friedrich J, Zhou P, Paninski L (2017) Fast online deconvolution of calcium imaging data. *PLoS Comput Biol* 13(3): e1005423. <https://doi.org/10.1371/journal.pcbi.1005423>

REVIEWER COMMENTS

Reviewer #1 (Remarks to the Author):

Christensen and Pillow provided a strong revision. The additional analyses and the new figures have solved most of my major concerns. There remains several minor points, mainly on the readability.

- 1) Explanation on new Fig. 2f-k is too sparse. The legend needs to describe more detail. Why are there clusters around 0.7 and -0.7 in many areas?
- 2) It remains unclear how many cells were included in each of the figure. For example, Figure 2c-d, how many cells are included in each of the bar graphs? How about the bar graphs in Supplementary Figure 1e? Please provide a table with N of data-points for each of the sub-figures.
- 3) The addition of the new Fig. 2 has ruined the figure referencing in the main text, impairing smooth reading.
- 4) Fig. 4a-d are more linked with Fig. 3 than with Fig. 4f-i. Reorganization of the subfigures would be helpful.
- 5) Readers usually expect to encounter the main figures in sequence. For instance, they expect explanations about Fig. 2b before those about Fig. 2f-k. Similarly for Fig. 4e.
- 6) In Methods, "integration time step = 0.05 ms, $C_m = 4.9$ ms, $g_l = .16$ us"
Units for membrane capacitance and leaky conductance cannot be ms or us. Capacitance should be F (farad), conductance should be S (Siemens)
- 7) P2 The last sentence of the first paragraph
"then when correlation....." should be "than when correlation...".
- 8) In the Discussion, please mention that the running behavior can induce activation in the limb somatosensory cortex, the limb motor cortex, probably auditory cortex (treadmill sound) and possibly activity related to whisking.

Reviewer #2 (Remarks to the Author):

In this revised manuscript, the authors take a look at the impact of locomotion on visual coding in several higher-order visual cortex areas in mouse. The authors have fixed a number of issues from the previous version, and some of the methods and goals have been clarified. There remain several issues that detract from the potential impact of this study.

Other than the initial analyses shown in the first figure, the authors have pooled neurons across all layers in their analysis of coding. Even in Figure 1, the proportions of positive, negative and non-monotonic correlations with locomotion are not separated out by layer. This is important for two reasons: 1) previous electrophysiology data suggests significant differences across layers in locomotion modulation, with the majority of layer 2/3 cells being positively modulated but <50% of layer 5 cells being positively modulated, and 2) pooling the different populations may obscure some of the changes across cortical areas, since not all populations project to higher-order areas. In addition, most previous imaging work has looked only at layer 2/3, and so much of the potentially novel insights here may be in the data from deeper layers.

The authors have added text on the caveats of GCaMP as a proxy for firing. However, there is an additional and possibly more relevant caveat, as decreases in activity are substantially more difficult to detect than increases. Thus, GCaMP data may not represent the full extent of decreased firing in cells with negative correlations between locomotion and firing rate as well as it does the increased firing in cells with positive correlations. As a minor side-note, 'noisy non-linear proxy for firing rate' is almost certainly an understatement, as the Allen dataset also includes simultaneous

patch/imaging data suggesting up to 90% of spikes are not faithfully reported by the GCaMP signal.

The analysis of behavioral states is troubling. A very wide swath of behavioral states are likely included in the '0 cm/s' locomotion condition, and this could muddy the outcome of the tuning and correlation analysis. There is no mention in the methods of how the pupil data were used. If the pupil correlation data in Supplementary Figure 1d is simply based on all the data, then it is essentially useless because locomotion and pupil are highly correlated, leading to very similar results as when locomotion is used. The authors should exclude all locomotion periods, as well as periods immediately before and after running, and run the correlation analysis using pupil data in the absence of running.

It is very unclear what the main thrust of the study is meant to be. On the one hand, there are potentially interesting findings about locomotion tuning and fidelity of visual encoding across the visual cortical hierarchy. On the other, there is modeling data about the impact of membrane voltage fluctuations on encoding. Either one of these would be fodder for a stand-alone paper, but in the current format there is insufficient development of either part. Indeed, the most interesting part of the current manuscript is the finding that even neurons with negative correlations with running may exhibit enhanced reliability. This could be enhanced with further analysis of data by layers, etc.

The authors may want to reference the extensive previous experimental and computational literature on the origins of membrane voltage fluctuations and their impact on encoding and transmission of information.

In figure 1b, the y axis label is F/dF , I assume the authors intended ' dF/F '

Reviewer #3 (Remarks to the Author):

In this resubmission, Christensen and Pillow examine GCaMP responses of neurons imaged in several visual cortical areas. They find that many cells' responses are suppressed by running and also by arousal as measured by pupil diameter. They find that decoder (logistic regression) performance increases while animals are running, as compared to when they are stationary. This effect holds true across the six visual areas they examined. With this revision, the conclusion remains weakly supported. While I commend the authors for their goals to understand coding via this dataset, the results are still fairly superficial and do not shed much light on how population activity in mouse visual cortex is coded.

Major:

- 1) As with comments from multiple reviewers in the earlier submission, it is essential when making statements like this -- on firing rate decreases and changes in variability with running -- to verify with data on spiking activity or at the very least have accurate calibrations of imaging responses. It is not sufficient to add a statement to the conclusion saying that their results could change if spikes are considered, as imaging nonlinearities are a big potential confound for this work. While the authors are now careful to frame their results as being results about the Allen Institute dataset, their conclusions must be about the information available in the brain, and not just about how analyses perform when applied to the dataset. In particular, the statements they make about variability (reliability) could well be a result of GCaMP imaging and might not hold up in spike recordings. Related: differences in results between dF/F measurements and deconvolved estimated spiking should be explored at least somewhat in the main figures.
- 2) The authors set up their finding that activity can be suppressed by running as fairly surprising and at odds with some views in the field. I find this result, while clear, to have limited impact in its present form. To support the conclusion they claim (disproving some common views in the field,

which they imply even if they are careful about what they say explicitly) about such suppression, it is essential to know more about the visual tuning properties of the neurons. Running effects have usually been previously analyzed in the field as a function of changes in tuned visual responses, and little is known in this dataset about the tuning properties of the neurons being analyzed (page 4, top). As one example, it could depend on the size of the stimuli used.

3) Model. While the authors do bound the conclusions they are drawing from their neuron simulations, the result does not add much insight; it is a fairly direct consequence of what is already known about influence of Vm variability in single LIF neurons, and there are a large number of network and neuron mechanisms that could produce such a result.

Minor

- p5: citations for fig 2 appear to be for fig 3 instead? Fig. 2: typos in title panel b, typos in axis labels panels f-k

- Fig 2: no title (general comment for other figures: recommend having each figure title be a declarative statement of the figure's result rather than a description of the analysis in the figure)

Reviewer #4 (Remarks to the Author):

The authors did a superb job replying to my the comments (and those of other reviewers), as well as and clarifying the language that was initially confusing. I have nor further comments.

We appreciate the opportunity to respond to a second round of reviewer comments. To address the points raised, we have completed new analyses of how recording depth / cortical layer impacts our results, and we have performed a new analysis to try to tease apart the contributions of running and pupil diameter. We have also re-organized the figures at the request of reviewer 1; we hope this new organization leads to a smoother reading experience. Finally, we have sought to clarify a variety of points and add appropriate caveats, as requested by the reviewers. A point by point response to the reviewers is contained below.

REVIEWER COMMENTS

Reviewer #1 (Remarks to the Author):

Christensen and Pillow provided a strong revision. The additional analyses and the new figures have solved most of my major concerns. There remains several minor points, mainly on the readability.

1) Explanation on new Fig. 2f-k is too sparse. The legend needs to describe more detail. Why are there clusters around 0.7 and -0.7 in many areas?

Well spotted! The clusters around ± 0.7 arise from the fact that these plots were created by selecting for neurons that are both “significantly tuned to running”, by the Levene's t-test (p value < 0.05), and responsive to the drifting grating stimulus. Selecting for visually responsive neurons was the main goal of this figure, and to be consistent with the rest of the paper we only include neurons with significant tuning to running. This selection for neurons that are significantly tuned to running tends to remove neurons with low correlation coefficients (i.e., the middle of the plot, creating the bands the reviewer noticed). We agree that the bands look a bit odd, so we have remade the figure without the selection criteria below. We have also included the selection criteria details in an updated main figure legend (“Plots include only data from neurons significantly tuned to running (Levene's t, $p < 0.05$), and responsive to drifting gratings”), and

included this new figure as a supplemental figure.

2) It remains unclear how many cells were included in each of the figure. For example, Figure 2c-d, how many cells are included in each of the bar graph? How about the bar graphs in Supplementary Figure 1e? Please provide a table with N of data-points for each of the sub-figures.

Sorry about that, we have now tried to make the information much easier to find. In general we have either added the information to the figure legends, or added an entire supplemental table when we thought that was a more appropriate way to convey the information. In the latter cases we have included notes in the relevant figure legends directing readers to the appropriate tables.

3) The addition of the new Fig. 2 has ruined the figure referencing in the main text, impairing smooth reading.

Oops very sorry about that, we fixed the issue.

4) Fig. 4a-d are more linked with Fig. 3 than with Fig. 4f-i. Reorganization of the subfigures would be helpful.

Done

5) Readers usually expect to encounter the main figures in sequence. For instance, they expect explanations about Fig. 2b before those about Fig. 2f-k. Similarly for Fig. 4e.

Thanks for the suggestion, we have reordered the figure panels.

6) In Methods, “integration time step = 0.05 ms, $C_m = 4.9$ ms, $g_l = .16$ us”

Units for membrane capacitance and leaky conductance cannot be ms or us. Capacitance should be F (farad), conductance should be S (Siemens)

We thank the reviewer for catching this oversight, and have corrected the units in the manuscript. $V_{\text{thresh}}: -49\text{mV}$, $V_{\text{init}}: -70$ mV, integration time step = 0.05 ms, $C_m = 490$ pF, $g_l = 16$ pS, $E_l = -65$ mV.

7) P2 The last sentence of the first paragraph

“then when correlation.....” should be “than when correlation...”.

Thanks again for your detailed edits, we’ve fixed this.

8) In the Discussion, please mention that the running behavior can induce activation in the limb somatosensory cortex, the limb motor cortex, probably auditory cortex (treadmill sound) and possibly activity related to whisking.

We appreciate this suggestion, and it makes sense, but we are concerned that it may not generally hold that those regions become more activated during running, so we are hesitant to include this sentence in the discussion. If one looks at wide-field imaging data of the entire mouse cortex (credit Tony Kim, Schnitzer lab) as a mouse increases its running speed, you can see that there is a relative decrease in activity in somatosensory cortex, and perhaps auditory cortex (it’s hard to see in these images as it falls over the edge of the field of view). These data are anecdotal, but we have seen similar trends in other published reports (e.g. Fisher, S. P. *et al.* Stereotypic wheel running decreases cortical activity in mice, *Nat. Commun.* 7, 13138 (2016)).

We mention this negative correlation to running in motor and somatosensory cortex in the context of arousal in the discussion, however the reviewer brings up a good point, and we have added the following caveat to that discussion.

However, it is also possible that the running induced negative correlation to running seen in somatosensory cortex and motor cortex (Fisher, S. P. *et al.* Stereotypic wheel running decreases cortical activity in mice. *Nat. Commun.* **7**, 13138 (2016)) are mechanistically distinct from what we see in some higher order visual cortices, as these regions are likely to be spuriously *activated* by any running related signals, such as interoception, whisking, etc. It's possible the negative correlation to running in motor and somatosensory signals instead acts to dampen these abundant, but perhaps not behaviorally useful signals.

Reviewer #2 (Remarks to the Author):

In this revised manuscript, the authors take a look at the impact of locomotion on visual coding in several higher-order visual cortex areas in mouse. The authors have fixed a number of issues from the previous version, and some of the methods and goals have been clarified. There remain several issues that detract from the potential impact of this study.

Other than the initial analyses shown in the first figure, the authors have pooled neurons across all layers in their analysis of coding. Even in Figure 1, the proportions of positive, negative and non-monotonic correlations with locomotion are not separated out by layer. This is important for two reasons: 1) previous electrophysiology data suggests significant differences across layers in locomotion modulation, with the majority of layer 2/3 cells being positively modulated but <50% of layer 5 cells being positively modulated, and 2) pooling the different populations may obscure some of the changes across cortical areas, since not all populations project to higher-order areas. In addition, most previous imaging work has looked only at layer 2/3, and so much of the potentially novel insights here may be in the data from deeper layers.

Thanks for this suggestion! We definitely agree this is a strength of the Allen Institute dataset. In addition to the initial laminar analysis we included (Figure 1c,d. Supplemental Figure 1f), we have broken out the analysis from figure 3 by layers, included below.

We find it intriguing that there seems to be a relatively large improvement in decoding performance in layer 4 in some of the regions in our study. We think this trend would be interesting to follow up on. A strong caveat on interpreting these data, however, are the differences in cell numbers and animal numbers between the different cre-lines. In particular there is far less data recorded in layer 4 than in layer 2/3. Thus we have opted to keep the laminar data in supplemental materials, as we feel it is not a part of our main narrative.

The authors have added text on the caveats of GCaMP as a proxy for firing. However, there is an additional and possibly more relevant caveat, as decreases in activity are substantially more difficult to detect than increases. Thus, GCaMP data may not represent the full extent of decreased firing in cells with negative correlations between locomotion and firing rate as well as it does the increased firing in cells with positive correlations. As a minor side-note, 'noisy non-linear proxy for firing rate' is almost certainly an understatement, as the Allen dataset also includes simultaneous patch/imaging data suggesting up to 90% of spikes are not faithfully reported by the GCaMP signal.

We agree with the reviewer, and have been following with interest the ongoing work comparing GCaMP to electrophysiology. We feel the main conclusion of this work is that calcium imaging and electrophysiology are simply two distinct types of measurement, each with their own biases. While calcium imaging is admittedly not great at detecting isolated spikes (in some circumstances—the 90% is something of a worst case measurement), electrophysiology has a strong bias towards high firing rate neurons (Siegle JH, Ledochowitsch P, Jia X, et al. Reconciling functional differences in populations of neurons recorded with two-photon imaging and electrophysiology. bioRxiv; 2020. DOI: 10.1101/2020.08.10.244723). Fundamentally, we agree that it's incorrect to think of calcium fluorescence imaging as a proxy for electrophysiology, and instead we think the two techniques should be considered independently, in light of their own strengths and weaknesses, as measurements of distinct physiological processes. We have changed our wording in the paper to reflect this. (See below)

An important caveat to our work is that although we have only measured and analyzed Ca²⁺ fluorescence as an indicator of neural activity, many previous theoretical studies on neural tuning, neural response gain, and even our own work on neural reliability through LIF simulations relied on analyzing neural firing rates. Ca²⁺ activity is a fundamentally different measure of neural activity and in many cases only bears a noisy, non-linear relationship to neural firing rates. Further work should be done to verify these findings with electrophysiology.

The analysis of behavioral states is troubling. A very wide swath of behavioral states are likely included in the '0 cm/s' locomotion condition, and this could muddy the outcome of the tuning and correlation analysis. There is no mention in the methods of how the pupil data were used. If the pupil correlation data in Supplementary Figure 1d is simply based on all the data, then it is essentially useless because locomotion and pupil are highly correlated, leading to very similar results as when locomotion is used. The authors should exclude all locomotion periods, as well as periods immediately before and after running, and run the correlation analysis using pupil data in the absence of running.

We appreciate the reviewer pushing us further in this direction! To try to address this concern we performed another control analysis, the figure is below.

In particular, we further subdivided trials when mice were stationary into trials when mice were stationary **and** had small pupils, or trials when mice were stationary **and** had large pupils. We compared decoding performance between these two conditions, and did not find a significant difference. There does seem to be a small trend towards improved performance during trials where the mice had dilated pupils, and it's very possible that if this analysis were performed in a dataset with more trials in this condition, a larger trend would emerge.

We have added this figure to the supplemental figures, and the following text to the decoding section:

We also performed a control analysis to try to separate the influence of arousal and locomotion, to determine whether a large variability in behavioral states underlying the “stationary” condition might contribute to poor decoding performance (Supplemental Figure 5). We did not find an effect of pupil diameter on decoding during the stationary periods, however it is possible that this is due to the relatively small number of trials included in this analysis, due to constraints of balancing conditions.

Supplemental Figure 5. a) distribution of pupil diameter during running and stationary periods. b) fraction of correctly classified trials comparing periods where mice were stationary and had dilated pupils (defined as pupil area greater than 3500 pixels, the pupil area which separates the running and stationary pupil distributions) and those stationary trials where mice had undilated pupils. Same as b. except broken out by region. In b. size of circle represents statistical effect size.

It is very unclear what the main thrust of the study is meant to be. On the one hand, there are potentially interesting findings about locomotion tuning and fidelity of visual encoding across the visual cortical hierarchy. On the other, there is modeling data about the impact of membrane voltage fluctuations on encoding. Either one of these would be fodder for a stand-alone paper, but in the current format there is insufficient development of either part. Indeed, the most interesting part of the current manuscript is the finding that even neurons with negative correlations with running may exhibit enhanced reliability. This could be enhanced with further analysis of data by layers, etc.

We apologize for the confusion, but we appreciate the reviewer finding so many aspects of our study interesting! We feel that our main finding is that negatively correlated neurons display enhanced reliability, however we think this finding is more interesting in the context of the overall correlation patterns of neural activity and running speed in the visual cortex. The LIF model is also intended to support the strength and interest of that main finding, by suggesting a possible mechanism for these results and to connect them to the previous literature.

The authors may want to reference the extensive previous experimental and computational literature on the origins of membrane voltage fluctuations and their impact on encoding and transmission of information.

Thank you for reminding us of this literature. We included several new sentences and citations in the Discussion:

Previous studies have suggested a variety of computational roles for membrane voltage fluctuations in visual cortex¹⁶, including aiding in contrast invariant tuning¹⁷. This raises the possibility that noise might be a general purpose computational tool that neurons leverage in order to respond flexibly to changing environments

In figure 1b, the y axis label is F/dF , I assume the authors intended ' dF/F '

Fixed, thank you.

Reviewer #3 (Remarks to the Author):

In this resubmission, Christensen and Pillow examine GCaMP responses of neurons imaged in several visual cortical areas. They find that many cells' responses are suppressed by running and also by arousal as measured by pupil diameter. They find that decoder (logistic regression) performance increases while animals are running, as compared to when they are stationary. This effect holds true across the six visual areas they examined. With this revision, the conclusion remains weakly supported. While I commend the authors for their goals to understand coding via this dataset, the results are still fairly superficial and do not shed much light on how population activity in mouse visual cortex is coded.

Major:

1) As with comments from multiple reviewers in the earlier submission, it is essential when making statements like this -- on firing rate decreases and changes in variability with running --

to verify with data on spiking activity or at the very least have accurate calibrations of imaging responses. It is not sufficient to add a statement to the conclusion saying that their results could change if spikes are considered, as imaging nonlinearities are a big potential confound for this work. While the authors are now careful to frame their results as being results about the Allen Institute dataset, their conclusions must be about the information available in the brain, and not just about how analyses perform when applied to the dataset. In particular, the statements they make about variability (reliability) could well be a result of GCaMP imaging and might not hold up in spike recordings. Related: differences in results between dF/F measurements and deconvolved estimated spiking should be explored at least somewhat in the main figures.

We appreciate the reviewer's concerns about the limitations of Ca²⁺ fluorescence imaging. In fact, the comparison between the two methodologies is extremely tricky, for many reasons brought up in the Allen institutes own paper on the matter (Siegle JH, Ledochowitsch P, Jia X, et al. Reconciling functional differences in populations of neurons recorded with two-photon imaging and electrophysiology. bioRxiv; 2020. DOI: 10.1101/2020.08.10.244723).

As they point out in this paper, ca²⁺ imaging is indeed a non-linear transformation of spikes (and, as the reviewer mentions, single spikes can be missed by calcium imaging). But, the important conclusion from the Siegle et. al. paper is the electrophysiology dataset itself is quite biased, missing low-firing rate neurons.

Importantly, supplemental figure 2 of the Siegel et. al. paper they note that in the electrophysiology dataset neurons tend to be less responsive, but more reliable during locomotion. This conclusion is consistent with our results, although it remains to be seen whether they also see these two effects in the same neurons, or just at a population level. Such a direct validation is beyond the scope of this work, which is fundamentally an analysis of calcium imaging data.

Finally, regarding deconvolution, this is a mistake in our previous reviewers response! We included a figure that was labeled "spearman's correlation coefficient" on the y-axis but should have been labelled "average firing rate". This figure did not show differences between deconvolved and non-deconvolved correlation to running, but instead showed an unrelated point that there was no clear relationship between correlation to running and event rate when deconvolving using oasis. The text was correct but the figure legend was incorrect (this mistake

was due to reusing code to generate the figure from our other analyses) — we apologize for the confusion! We have included the corrected figure below.

2) The authors set up their finding that activity can be suppressed by running as fairly surprising and at odds with some views in the field. I find this result, while clear, to have limited impact in its present form. To support the conclusion they claim (disproving some common views in the field, which they imply even if they are careful about what they say explicitly) about such suppression, it is essential to know more about the visual tuning properties of the neurons. Running effects have usually been previously analyzed in the field as a function of changes in tuned visual responses, and little is known in this dataset about the tuning properties of the neurons being analyzed (page 4, top). As one example, it could depend on the size of the stimuli used.

We politely disagree with the reviewer on this point. The initial running modulation paper by Niell & Stryker (Neuron, 2010) has nearly a 1000 citations and specifically mentions the increase in firing rate they see is independent of tuning properties. We quote their abstract below:

“Most neurons showed more than a doubling of visually evoked firing rate as the animal transitioned from standing still to running, without changes in spontaneous firing or stimulus selectivity.” Niell and Stryker, *Neuron*, 2010

Similarly, the texts of most papers we looked at comment on how running is accompanied by dramatic changes in firing rates, and how it increases the gain of visual responses — without mentioning changed tuning properties. These two further examples (including from a review) are, to us, indicative of how the field broadly views the relationship between locomotion and neural activity in V1.

“During locomotion, V1 neurons in layers 2/3 and 4 have more depolarized membrane potentials [2, 3], higher firing rates [2, 3, 4, 5, 6], and increased tuning gain [2, 4].” (Sinem Erisken, Agne Vaiceliunaite, Ovidiu Jurjut, Matilde Fiorini, Steffen Katzner, Laura Busse, Effects of Locomotion Extend throughout the Mouse Early Visual System, *Current Biology*, 2014)

“Behavioural state has a strong influence on cortical processing [12,13]. For instance, visual responses in V1 are stronger, more reliable, and less correlated when mice walk or run compared to when they are quietly resting [14, 15, 16]. These effects show similarities to modulation of responses by arousal or attention [9,17, 18, 19, 20].” (Adil G Khan, Sonja B Hofer, Contextual signals in visual cortex, *Current Opinion in Neurobiology*, Volume 52, 2018)

We think investigating this issue with greater nuance is extremely important, for example as a function of size of stimuli (e.g. Asl̄Ayaz, Aman B. Saleem, Marieke L. Schölvinck, Matteo Carandini, “Locomotion Controls Spatial Integration in Mouse Visual Cortex”, *Current Biology*, 2013).

In our work, all the drifting gratings were full-field, so we are only comparing one level of surround-suppression. It is interesting to note that, if we understand the above paper correctly, this is where we might see the most difference between running and stationary periods. We will add a caveat (reproduced below) to the discussion that our results might change if different sizes of visual stimuli were compared.

For example, previous work ⁶ has shown that the response to locomotion is highly dependent on stimuli size -- thus our results might change if we tested a variety of non-full field stimuli.

3) Model. While the authors do bound the conclusions they are drawing from their neuron simulations, the result does not add much insight; it is a fairly direct consequence of what is already known about influence of Vm variability in single LIF neurons, and there are a large number of network and neuron mechanisms that could produce such a result.

The point of the model was to provide a sufficient explanation for our results in terms of previously observed neural phenomenon during arousal in visual cortex (Martin Vinck, Renata Batista-Brito, Ulf Knoblich, Jessica A. Cardin, Arousal and Locomotion Make Distinct Contributions to Cortical Activity Patterns and Visual Encoding, *Neuron*, 2015)) as a potential mechanism for our results. We feel this will be helpful for some readers in providing intuition for at least one plausible mechanism that could give rise to the observed decoding phenomenon. However, we agree that it is just one possible explanation, and that other network and single-neuron mechanisms are possible. The following passage in the LIF section is intended to make this point clear:

There are many important features of actual cortical circuits (such as recurrence between visual areas, and even the fact that higher order visual areas receive input

from lower order visual regions, which may already have running speed modulated activity), which are not included in these simulations. These simulations are intended to show that all else being equal, simply the changing background noise can cause all of the effects we observe in our data (e.g. would, in the absence of other factors, both decrease firing rates and increase response reliability).

Minor

- p5: citations for fig 2 appear to be for fig 3 instead? Fig. 2: typos in title panel b, typos in axis labels panels f-k

Fixed, thank you!

- Fig 2: no title (general comment for other figures: recommend having each figure title be a declarative statement of the figure's result rather than a description of the analysis in the figure)

Done, thank you.

Reviewer #4 (Remarks to the Author):

The authors did a superb job replying to my comments (and those of other reviewers), as well as clarifying the language that was initially confusing. I have no further comments.

We thank the reviewer for reading our manuscript, and for their helpful comments!!

REVIEWER COMMENTS

Reviewer #2 (Remarks to the Author):

Authors have fully addressed my concerns.

Reviewer #3 (Remarks to the Author):

This is a resubmission of a manuscript by Christensen and Pillow that analyzes data from the Allen Institute.

In my view the data shown still does not provide convincing support for their main conclusions.

TUNING, GAIN, AND VISUAL RESPONSES: The core of the paper is (p2) is that "[the work] quantified how running speed affects visual responses in six visual cortical regions." But the data shown does not characterize this -- it does not go into sufficient depth about visual responses and interactions with running speed -- and so it is impossible to determine whether this is a novel effect. For example, the foundational finding about running effects in V1 (Niell and Stryker 2010) was about changes in visually evoked firing in V1. From their abstract: "Most neurons showed more than a doubling of visually-evoked firing rate as the animal transitioned from standing still to running, without changes in spontaneous firing or stimulus selectivity." That is, there is a change in response gain. Yet the current work mixes changes in spontaneous activity and changes in evoked activity, without ever providing clear characterization of the visual response tuning of the neurons. Fig 2, added in a revision, shows no correlation between running-response correlation (that is, gain!) and tuning in V1. How should we interpret this? Responsive but unselective cells still show gain changes with running? Perhaps, but then why is responsivity not plotted? How are population responses (PSTHs) changed by running speed? The text does not even explain what are the stimuli shown to the animals. Supp Fig. 1a shows fraction of neurons from model fits without cell examples or population plots. Even the few examples shown in Fig. 1b are not obviously labeled in figure or legend as to what area they come from.

Prior studies of visual neurons perform analyses like this in part to rule out artifacts, which can be numerous: neuropil issues, saturation due to GCaMP, motor artifacts that correlate with running that may drive visual neurons, etc, etc.

In sum, the data shown do not fully characterize the response properties of these cells, have features that may or may not conflict with the previous literature, and do not seem self-consistent. For these reasons I cannot assess whether the changes with running seen in the data are truly new and novel effects or can be explained by other artifactual effects. This issue is similar to what I interpret R2 to mean in last round of reviews when he/she says "[I]n the current format there is insufficient development [of the running effects]."

I apologize for any confusion in my previous review, where I asked for more characterization of visual properties, and the authors rebutted instead the suggestion that running changed tuning. I see how my "changes in tuned visual responses" could have been misinterpreted. My intended meaning was "changes in visual responses at different points on the tuning function [to assess potential changes in gain]."

GCaMP: The rebuttal did not address my key point: reliability and variability can be distorted by calcium indicators, for example by saturation. This potentially confounds the decoding analysis. There is no redline and the author's response does not convey if anything was added to the paper on this, so it is difficult for me to check what was added to the manuscript, but it should have been essential that this be addressed in the main text.

MODEL: This part of paper is fine, but still reflects just one of many potential mechanisms, without evidence it is the one used.

WRITING: this is a short paper and it does not carry enough information to explain what was done.

For example, in Fig. 1, the visual stimuli used and experimental setup are not clearly described. Yes, this is a re-analysis, but such a description is absolutely fundamental to the results. And the writing is confusing at other points. For example, this statement is very unclear: "We think this story is a cautionary tale for interpreting the 'improved coding' in V1 during locomotion in an ethological way with respect to locomotion per-se, as it's quite possible that the locomotion specific changes in V1 are not responsible for the overall improved coding, instead this improvement could be due to a mechanistically and behaviorally distinct mechanism."

We thank the reviewer for their thoughtful comments, which helped us to improve our paper, both through new analysis and narrative reorganization. In particular we have included new analysis of how population tuning curves change between periods of locomotion and quiescence. This new analysis is now Figure 3, and we believe helps tie together the tuning and decoding pieces of our work. In addition, we have rewritten and reorganized the entire manuscript for increased clarity and flow. We appreciate the opportunity to revise our manuscript, and believe these changes have substantially improved the paper.

REVIEWER COMMENTS

Reviewer #3 (Remarks to the Author):

This is a resubmission of a manuscript by Christensen and Pillow that analyzes data from the Allen Institute.

In my view the data shown still does not provide convincing support for their main conclusions.

TUNING, GAIN, AND VISUAL RESPONSES: The core of the paper is (p2) is that "[the work] quantified how running speed affects visual responses in six visual cortical regions." But the data shown does not characterize this -- it does not go into sufficient depth about visual responses and interactions with running speed -- and so it is impossible to determine whether this is a novel effect. For example, the foundational finding about running effects in V1 (Niell and Stryker 2010) was about changes in `_visually evoked_` firing in V1.

From their abstract: "Most neurons showed more than a doubling of visually-evoked firing rate as the animal transitioned from standing still to running, without changes in spontaneous firing or stimulus selectivity." That is, there is a change in response gain. Yet the current work mixes changes in spontaneous activity and changes in evoked activity, without ever providing clear characterization of the visual response tuning of the neurons.

We thank the reviewer for pushing us on this point. We apologize if we were not clear in the previous submission, but we did not mix together changes in spontaneous activity and changes in evoked activity. In Figure 1, we did indeed show neural correlations to running speed, averaged across all stimuli. However, contrary to the reviewers comment, we **never** mixed changes in spontaneous activity with changes in evoked activity. In sub-panels 1D and

1F of Figure 1, we compared spontaneous activity during running and stationary periods. For the rest of the paper, however, we used stimulus-evoked responses only.

We did the analysis in Figure 1 under the assumption (as has been shown in many previous works, including e.g., Neill and Stryker 2010) that locomotion does not change the preferred stimulus identity of neurons. Therefore, averaging responses across all the stimuli presented gives us higher statistical power to assess changes in evoked activity levels (regardless of what stimulus is presented) during locomotion. Also, we would like to note that **all analyses** other than Figure 1 assessed neural activity of visually responsive neurons only during the drifting gratings stimulus.

As an aside, unlike Neill and Stryker, we *did* see changes in spontaneous activity during locomotion. Ours is not the only subsequent work to have seen such changes, e.g. Saleem et. al. [1] say “We found that most V1 neurons responded to locomotion even in the dark.” (e.g. spontaneous activity).

With the addition of Figure 2 in the last rebuttal, we analyzed the correlation between evoked responses and running separately during the preferred and non-preferred drifting grating stimuli for each neuron. We believe this assesses “gain” exactly as requested by the reviewer, although we apologize if we have misunderstood the reviewer’s request.

In order to dig deeper into this issue, and the reviewers' suggestion to assess population tuning curves, we have reworked figure 2 of our manuscript (which is now figure 3). This figure shows a new population tuning curve analysis which we think demonstrates our core point — that reliability increases even in neurons whose overall activity does not increase during

locomotion very cleanly. The new figure and associated text is included below.

Figure 3 Increased reliability accounts for decoding results . a. Population tuning curves separately plotted for neurons that have a negative correlation coefficient with running speed, and those that have a positive correlation tuning curve. Neural responses were z-scored across all stimuli before being split into running and stationary groups, then sorting according to their average response to each stimuli. Δff zscore is averaged across all cells. b) histogram comparing the average change in neural activity for the preferred stimuli, vs. all other stimuli. Percent difference is averaged across all cells in a particular imaging dataset before plotting. A positive value corresponds to cases where the average response is higher during running, and a negative value corresponds to cases where the average response is lower during running. c. reliability as defined in panel “d”, averaged across all neurons per mouse-dataset, in running vs. stationary periods. d. Schematic of reliability calculation. e. scatter plots of variance of average responses and variance of individual responses – the numerator and denominator of the

reliability metric, respectively. F. Correlation between decoding performance and average percent change in reliability in each experiment. G. Same as a, b except decoders trained excluding top 25% of cells with the most changed reliability. H. Same as a, I. except decoders trained excluding top 50% of cells with the most changed reliability.

[...]When we examined population tuning curves (Figure 3 a), we found that even in cells with negative overall running-activity correlations, locomotion tended to **increase** the response to the most preferred stimuli, while the activity across all the other stimuli decreased (Figure 3b). This was consistent across each individual region we examined other than posterior medial visual cortex, where the activity to the preferred stimulus did not increase (Supplemental Figure. 8c). Thus, in tuned cells, even a net neural activity decrease during locomotion might increase the signal to noise ratio, and thus the decodability of visual stimuli. In these data a selective reduction in background activity (aka activity to non-preferred stimuli, but not a reduction in overall response noise) is the primary cause of the overall activity reduction we previously observed, when marginalizing across all stimuli Figure 3e.

Indeed, we found that even neurons whose activity did not increase during running showed increased 'reliability', defined as the variance of each neuron's average response to each image divided by the total variance of each neuron's response¹⁰, and this improved reliability was correlated with the increased decoding performance (Fig 3c, f). We excluded the top 25% and top 50% of neurons whose reliability changed in each separate experimental session, and re-trained and tested decoders. We found that, although the decoders still performed similarly to decoders trained with all neurons, with 25% of neurons excluded, the difference between running and stationary decoding accuracy was greatly diminished, and it was abolished when the top 50% of neurons with changed reliability were excluded (Fig 3e,f). Thus, unlike previous reports based on data from V1, we found increased decoding accuracy correlated with increased fidelity in single neuron responses, and was not explainable entirely by decreased noise correlations or increased overall activity levels.

Fig 2, added in a revision, shows no correlation between running-response correlation (that is, gain!) and tuning in V1. How should we interpret this? Responsive but unselective cells still show gain changes with running? Perhaps, but then why is responsivity not plotted? How are population responses (PSTHs) changed by running speed?

We apologize for the confusion about this plot. We have completely revamped the old figure 2 (and rearranged the figures, this analysis is now in supplemental figure 8, shown below). As requested by the reviewer, we now show the distribution of correlation coefficients to running separately between responsive and non responsive neurons, across all regions. Our point is simply that even if one only considers responsive neurons, there is still a distribution of correlation between running and neural activity, including many negatively modulated cells. The new supplemental figure (and it's associated main text figure, above) also includes the population PSTH analysis suggested by the reviewer. The supplemental figure and it's related text are reproduced below.

Supplemental Figure 8. A) average correlation coefficient between running and neural activity, grouped by responsiveness to drifting gratings. A neuron is considered responsive to drifting gratings if it passes an Anova with $p < 0.05$ comparing the distribution of responses to all stimuli, including the blank stimuli, with the null hypothesis that all distributions are the same. B) the histogram density of running speed – neural activity correlation coefficients, split up by region and responsiveness to the drifting gratings stimuli. C) Population tuning curves comparing the average neural response to each drifting grating direction, sorted from most to least preferred (right to left on the x axis of all plots). Neural activity was z-scored across all stimuli responses prior to splitting into running / not-running conditions.

Motivated by these findings, we hypothesized that individual neurons might encode the stimulus identity more reliably when the animal was running, even if their overall activity

did not increase. To assess the relationship between stimulus encoding and running-speed correlation, we restricted our correlation analysis to the same drifting gratings stimulus used for our decoding analysis. We first examined the distribution of running speed – activity correlations across the visual regions, similarly to our analysis from figure 1. We observed that, although in general drifting gratings responsive neurons tend to be more positively correlated with running than drifting gratings non-responsive neurons (Supplemental Figure 8a,b), neurons in all regions showed a diversity of running speed – activity correlations.

The text does not even explain what are the stimuli shown to the animals. Supp Fig. 1a shows fraction of neurons from model fits without cell examples or population plots. Even the few examples shown in Fig. 1b are not obviously labeled in figure or legend as to what area they come from.

Thanks for pointing out this oversight. We have updated the figure legend of Fig1b. to indicate these cells were from layer 2/3 of primary visual cortex. We have also included the following new supplemental figure with examples of our model fits. Our text now explicitly states which stimulus is shown to the animals for each experiment.

Supplemental Figure 2. Examples of cells whose activity are best fit by increasing, decreasing, and band-pass gaussian models, respectively. In each plot the average neural activity (in black), and all fit models (yellow, pink, and blue) are shown. Plotted neural data are from a held-out test set, model fits are the MLE model from the training set. Best fit model was determined by lowest residual on the test set, across 10 cross-validation folds. All cells are from layer 2/3 of primary visual cortex. Models were fit on data collected while any of the visual stimuli were presented.

Prior studies of visual neurons perform analyses like this in part to rule out artifacts, which can be numerous: neuropil issues, saturation due to GCaMP, motor artifacts that correlate with running that may drive visual neurons, etc, etc.

Neuropil: Standard neuropil correction was applied. From the allen brain observatory white paper: "Neuropil Subtraction. The recorded fluorescence from an ROI was contaminated by the fluorescence of the neuropil immediately above and below the cell due to the point-spread function of the microscope. In order to correct for this contamination, the amount of contamination was estimated for each ROI. The estimated $\square\square$ was done by taking an annulus of 10 μm around the cellular ROI, excluding pixels from any other ROIs. In order to remove this contamination, the extent to which ROI was affected by its local neuropil signal was evaluated." In our methods section we mention the neuropil correction was performed, and direct readers to the Allen Institute white paper for details.

GCaMP non-linearity: we have added a caveat to the discussion specifically about gcamp non-linearities.

A motion correction algorithm was run on the two-photon movies prior to cell extraction. Movies were also manually checked for excessive motion artifacts.

In sum, the data shown do not fully characterize the response properties of these cells, have features that may or may not conflict with the previous literature, and do not seem self-consistent. For these reasons I cannot assess whether the changes with running seen in the data are truly new and novel effects or can be explained by other artifactual effects. This issue is similar to what I interpret R2 to mean in last round of reviews when he/she says "[I]n the current format there is insufficient development [of the running effects]."

We have tried to clarify our analysis, and hope this helps the reviewer evaluate our work. We have included a reorganized manuscript with track-changes enabled, as well as the text and figure of our reworked characterization of the response properties of the cells.

I apologize for any confusion in my previous review, where I asked for more characterization of visual properties, and the authors rebutted instead the suggestion that running changed tuning. I see how my "changes in tuned visual responses" could have been misinterpreted. My intended meaning was "changes in visual responses at different points on the tuning function [to assess potential changes in gain]."

We thank the reviewer for this clarification. We have done exactly that in our previous Figure 2. We assessed neural responses to preferred and non-preferred stimuli, during periods of quiescence and locomotion. In the current manuscript we have extended this analysis to show responses separated across all grating orientations.

We think our reworked Figure 2(3) illustrates nicely the changes in population tuning curves during locomotion. This figure is consistent with the idea that during locomotion/arousal, the population tuning curves SNR is increased by lowering the response to the non-preferred stimuli, as well as by driving increased responsivity to preferred stimuli. The entire new figure and associated text is included in the response to the first comment.

GCAMP: The rebuttal did not address my key point: reliability and variability can be distorted by calcium indicators, for example by saturation. This potentially confounds the decoding analysis. There is no redline and the author's response does not convey if anything was added to the paper on this, so it is difficult for me to check what was added to the manuscript, but it should have been essential that this be addressed in the main text.

In the previous revision, we sought to address the reviewer's point about reliability and variability by adding caveats about GCaMP in the Discussion section. We apologize if these did not go far enough in addressing the reviewer's point. We have now have added further explicit commentary on how the non-linearity of calcium indicators could impact our results. Text is included below!

"An important caveat to our work is that although we have only measured and analyzed Ca²⁺ fluorescence as a proxy for neural activity, many previous theoretical studies on neural tuning, neural response gain, and even our own work on neural

reliability through LIF simulations relied on analyzing neural firing rates. Ca²⁺ activity is a fundamentally different measure of neural activity and in many cases only bears a noisy, non-linear relationship to neural firing rates. In particular Ca²⁺ sensors display a non-linear activity / fluorescence relationship, which could potentially impact analysis of neural response reliability, and overall correlation to running. Further work should be done to verify these findings with electrophysiology.”

MODEL: This part of paper is fine, but still reflects just one of many potential mechanisms, without evidence it is the one used.

WRITING: this is a short paper and it does not carry enough information to explain what was done. For example, in Fig. 1, the visual stimuli used and experimental setup are not clearly described. Yes, this is a re-analysis, but such a description is absolutely fundamental to the results. And the writing is confusing at other points. For example, this statement is very unclear: "We think this story is a cautionary tale for interpreting the 'improved coding' in V1 during locomotion in an ethological way with respect to locomotion per-se, as it's quite possible that the locomotion specific changes in V1 are not responsible for the overall improved coding, instead this improvement could be due to a mechanistically and behaviorally distinct mechanism."

We have edited our manuscript to increase clarity. The edited document is included with changes tracked. We hope this addresses the reviewers' concerns. In particular we streamlined the discussion section to improve clarity/simplicity, and remove speculative comments such as the one above. Our streamlined discussion section is reproduced below.

In conclusion, we observed a striking diversity in the type of neural activity changes during locomotion across different visual cortical regions. Surprisingly, neurons in some higher order visual areas were more likely to be negatively correlated with running than positively correlated with running, in contrast to previous results showing response enhancement V1. This negative correlation is not easily

reconcilable with theories that explain the running speed modulation in V1 simply by enhanced 'attention' to vision during

However, despite the trend towards running-induced suppression in higher order visual cortices, running still enhanced the representations of visual stimuli, as measured by decoding accuracy. This effect was not attributable to noise correlations or increased magnitude of neural activity, but instead was attributable to increased reliability of individual neural responses.

Physiologically, both the decrease in activity we observed, and the increase in reliability could result from lower background membrane voltage fluctuations during locomotion⁸. Considering that activity in somatosensory and motor cortex are also negatively correlated with running¹², it is possible that this negative correlation we observe is a more generic cortical phenomenon. However, it is also possible that the running induced negative correlation to running seen in somatosensory cortex and motor cortex¹² are mechanistically distinct from what we see in some higher order visual cortices, as these regions are likely to be spuriously activated by any running related signals, such as interoception, whisking, etc. It's possible the negative correlation to running in motor and somatosensory signals instead acts to dampen these abundant signals.

Alternately, given that the correlation with running speed seems to be dependent on how strongly a neuron is driven by a particular stimulus, it's possible that as a community we are just better at selecting stimuli that specifically drive V1 neurons than almost any other region, and thus V1 is the region in which stimulus-gain effects are the most obvious.

Similarly, our results could be a downstream consequence of a difference in distribution of preferred stimuli type across the different visual regions – as previous authors have noticed that neurons with particular tuning preferences (namely those tuned to high spatial frequencies¹⁶) are more likely to be positively correlated with running. This effect has been described as a potential rodent analog

of spatial attention. However this is unlikely to completely explain our results -- the published data on spatial frequency selectivity in mouse higher order visual cortex¹⁴ does not correlate with the distribution of running speed selectivity we have observed. For example in Marshel et. al. AL has the lowest spatial selectivity, and PM has the highest spatial selectivity, whereas in our data AL and PM have relatively similar overall distributions of neural correlations with locomotion speed.

An important caveat to our work is that although we have only measured and analyzed Ca²⁺ fluorescence as a proxy for neural activity, many previous theoretical studies on neural tuning, neural response gain, and even our own work on neural reliability through LIF simulations relied on analyzing neural firing rates. Ca²⁺ activity is a fundamentally different measure of neural activity and in many cases only bears a noisy, non-linear relationship to neural firing rates. In particular Ca²⁺ sensors display a non-linear activity / fluorescence relationship, which could potentially impact analysis of neural response reliability, and overall correlation to running. Further work should be done to verify these findings with electrophysiology.

In either case, these results highlight previously unknown differences in running speed modulation in primary visual cortex and posterior and anterior higher order visual cortices, and highlight important questions for future work.

1] Saleem, A., Ayaz, A., Jeffery, K. *et al.* Integration of visual motion and locomotion in mouse visual cortex. *Nat Neurosci* 16, 1864–1869 (2013). <https://doi.org/10.1038/nn.3567>

REVIEWERS' COMMENTS

Reviewer #3 (Remarks to the Author):

I appreciate that the authors have worked to revise this manuscript to address my comments. Thank you for the effort.

I will add a few comments on the latest rebuttal letter, and I hope the authors interpret these constructively as just ways this type of work could be improved.

- "Averaging responses across all the stimuli gives us higher statistical power..." Yes, I agree, but this statement confirms the issue. Tuning curve changes were not well-characterized. And these are essential to relate to the literature.

- I don't think it's a good idea to re-sort all stimulus levels per neuron (Fig. 3a.) to characterize tuning changes. This isn't a tuning curve. While I'd like to see how tuning curve width and preferred directions, etc might change, I do appreciate the effort: his new panel does bring some new information for readers.

- The GCaMP saturation issue is important and is now mentioned only in passing. Saturation has a clear effect on population decoding that should be analyzable, and it should be possible to explicitly discuss how this would affect the responses.

We appreciate the reviewers time and attention to our manuscript. Below are our responses to their comments on our most recent revision, in line with the reviewer's comments.

I appreciate that the authors have worked to revise this manuscript to address my comments. Thank you for the effort.

I will add a few comments on the latest rebuttal letter, and I hope the authors interpret these constructively as just ways this type of work could be improved.

- "Averaging responses across all the stimuli gives us higher statistical power..." Yes, I agree, but this statement confirms the issue. Tuning curve changes were not well-characterized. And these are essential to relate to the literature.

We appreciate the reviewers comment, and definitely agree -- however, as we commented in previous responses, this is a limitation of the **dataset**, not our analysis. The dataset includes only a few repetitions of each stimulus, and when we split by running and stationary periods, the tuning curves are far too noisy to be interpreted. We hope our work inspires future experiments more well suited to detailed tuning curve analysis.

- I don't think it's a good idea to re-sort all stimulus levels per neuron (Fig. 3a.) to characterize tuning changes. This isn't a tuning curve. While I'd like to see how tuning curve width and preferred directions, etc might change, I do appreciate the effort: his new panel does bring some new information for readers.

We're glad the reviewer appreciates the effort, and finds it informative! We'd like to make individual neuron tuning curves, but as we commented above they are too noisy to be informative.

- The GCaMP saturation issue is important and is now mentioned only in passing. Saturation has a clear effect on population decoding that should be analyzable, and it should be possible to explicitly discuss how this would affect the responses.

We thank the reviewer for their comment.

In the df/f curves presented in Dana et. al. GCaMP6 saturates above ~ 80 hz, and below ~ 1 hz. In Niell and Stryker fig 3b, the median evoked firing rate (the most relevant value for most of our experiments, as it is representative of response across all stimuli) was between ~ 3 hz and ~ 8 hz, with a maximum evoked firing rate of 30 hz. According to these data the saturation at low firing rates is more of a concern than saturation at high firing rates. That is, we have less sensitivity to identify neurons whose median evoked activity **decreases** during locomotion than we have to identify those whose activity **increases** - thus our results about neurons whose activity are reduced during running are unlikely to be caused by gcamp saturation.

By the same argument, our results about reliability are unlikely to be caused due to gcamp saturation, as we show a decrease in the lowest evoked responses, but no decrease of the highest evoked responses. Nonetheless we appreciate the reviewers' concerns, and will further strengthen our GCaMP caveat. The original and strengthened versions are included below.

original:

An important caveat to our work is that although we have only measured and analyzed Ca²⁺ fluorescence as a proxy for neural activity, many previous theoretical studies on neural tuning, neural response gain, and even our own work on neural reliability through LIF simulations relied on analyzing neural firing rates. Ca²⁺ activity is a fundamentally different measure of neural activity and in many cases only bears a noisy, non-linear relationship to neural firing rates. In particular Ca²⁺ sensors display a non-linear activity / fluorescence relationship, which could potentially impact analysis of neural response reliability, and overall correlation to running. Further work should be done to verify these findings with electrophysiology.

updated:

An important caveat to our work is that although we have only measured and analyzed Ca²⁺ fluorescence as a proxy for neural activity, many previous theoretical studies on neural tuning, neural response gain, and even our own work on neural reliability through LIF simulations relied on analyzing neural firing rates. Ca²⁺ activity is a fundamentally different measure of neural activity and in many cases only bears a noisy, non-linear relationship to neural firing rates. In particular Ca²⁺ sensors display a non-linear activity / fluorescence relationship, **with saturation both at high and low firing rates**, which could potentially impact analysis of neural response reliability, and overall correlation to running. **In particular, small changes in neural activity when firing rates are low can be difficult to detect with GCaMP. This could limit our ability to detect decreases in spontaneous activity, and decreases in activity evoked in response to presentation of non-preferred stimuli.** Further work should be done to verify these findings with electrophysiology.

Dana, H., Sun, Y., Mohar, B. *et al.* High-performance calcium sensors for imaging activity in neuronal populations and microcompartments. *Nat Methods* 16, 649–657 (2019).
<https://doi.org/10.1038/s41592-019-0435-6>